# Hydrogel tapes for fault-tolerant strong wet adhesion

Bin Xue [1,6], Jie Gu[1,6], Lan Li[2,6], Wenting Yu[1], Sheng Yin[1], Meng Qin[1], Qing Jiang[2✉], Wei Wang [1,3✉] & Yi Cao [1,3,4,5✉]

Fast and strong bio-adhesives are in high demand for many biomedical applications, including closing wounds in surgeries, fixing implantable devices, and haemostasis. However, most strong bio-adhesives rely on the instant formation of irreversible covalent crosslinks to provide strong surface binding. Repositioning misplaced adhesives during surgical operations may cause severe secondary damage to tissues. Here, we report hydrogel tapes that can form strong physical interactions with tissues in seconds and gradually form covalent bonds in hours. This timescale-dependent adhesion mechanism allows instant and robust wet adhesion to be combined with fault-tolerant convenient surgical operations. Specifically, inspired by the catechol chemistry discovered in mussel foot proteins, we develop an electrical oxidation approach to controllably oxidize catechol to catecholquinone, which reacts slowly with amino groups on the tissue surface. We demonstrate that the tapes show fast and reversible adhesion at the initial stage and ultrastrong adhesion after the formation of covalent linkages over hours for various tissues and electronic devices. Given that the hydrogel tapes are biocompatible, easy to use, and robust for bio-adhesion, we anticipate that they may find broad biomedical and clinical applications.

[1] Collaborative Innovation Center of Advanced Microstructures, National Laboratory of Solid State Microstructure, Key Laboratory of Intelligent Optical Sensing and Manipulation, Ministry of Education, Department of Physics, Nanjing University, 210093 Nanjing, China. [2] State Key Laboratory of Pharmaceutical Biotechnology, Division of Sports Medicine and Adult Reconstructive Surgery, Department of Orthopedic Surgery, Drum Tower Hospital affiliated to Medical School of Nanjing University, 210008 Nanjing, China. [3] Institute for Brain Sciences, Nanjing University, 210093 Nanjing, China. [4] Chemistry and Biomedicine Innovation Center, Nanjing University, 210093 Nanjing, China. [5] Wenzhou Institute, University of Chinese Academy of Sciences, 325001 Wenzhou, China. [6] These authors contributed equally: Bin Xue, Jie Gu, Lan Li. ✉email: qingj@nju.edu.cn; wangwei@nju.edu.cn; caoyi@nju.edu.cn

Sutures have long been the first choice in surgery for hemostasis, wound closure, and integration of bioelectronic devices[1,2]. However, suturing may cause tissue damage or scars at the fixing points and is difficult to apply to tissues with complex structures (e.g., spinal cord and heart). Recently, bio-adhesives have emerged as potential alternatives to sutures[3–6], as they are biocompatible, nontoxic, and easy to use[7,8]. Depending on the formulation, bio-adhesives can be either glue-type[9–20] or tape-type[21–23] adhesives. Glue-type adhesives require a long curing process (hours to days) to establish strong cohesion and interfacial adhesion. This long curing time greatly limits their applications in scenarios that require rapid and strong adhesion, such as attaching bioelectric devices to a beating heart or instant hemostasis[17,18]. Tape-type adhesives can be instant and reversible, but the adhesion strength is typically low (<50 kPa) if they only form non-covalent bonds with the tissues. They are also prone to detach from surfaces due to swelling of the adhesives or bleeding from the tissue[24–28]. To circumvent these drawbacks, tape-type adhesives based on covalent surface bonding have been introduced[29–31]. These adhesives are mechanically robust and can function properly on dynamic surfaces or tissues bearing considerable tension. However, once covalent adhesion bonds are formed, the adhesives cannot be removed easily[32–34]. In particular, as most surgeries are operated on dynamic tissue surfaces, mispositioning tape-type adhesives is almost unavoidable. It remains challenging to develop bio-adhesives that can rapidly, robustly, and conformally integrate with various wet tissues or bioelectronic devices yet detachable if misplaced[35]. To meet these needs, a few strategies have been introduced. For example, Suo and co-workers implemented a photolysable bond to allow detachment of adhesives upon UV light illumination[36]. Zhao and co-workers introduced a disulfide bond into covalent linkages to render redox-controlled reversible detachment and attachment[37]. Despite the great success of these methods, using additional triggers still makes detachment inconvenient. Moreover, the chemical reactions used for detachment are often slow and not fully reversible.

In nature, marine organisms, such as mussels, can strongly bind to various organic and inorganic surfaces by secreting various proteaceous adhesives containing dihydroxyl phenylalanine (dopa)[38–40]. More importantly, they have evolved a special mechanism to control the adhesion strength at different timescales. In freshly secreted adhesive proteins, the catechol group of dopa forms non-covalent interactions with various substrates (i.e., charge–charge interactions, π–π interactions, cation–π interactions, coordination, hydrogen bonding, and hydrophobic effects)[15,16,41]. Then dopa is gradually oxidized to dopaquinone and forms hydrogen bonds with unoxidized catechol groups as well as amino or carboxyl groups[42]. The aggregation of hydrophobic dopaquinone provides additional cohesion. Eventually, dopaquinone forms covalent linkages with primary amine and thiol groups slowly (in hours) to render strong adhesion[43–46]. This timescale-dependent adhesion is precisely regulated by the pH of the environment and the presence of enzymes[47]. Remarkably, through this complicated dopa chemistry, mussels can integrate simultaneous strong and instant adhesion with a fail-safe timing mechanism to prevent premature solidification in the secretory system or mistargeted adhesion. However, in many synthetic adhesives with catechol-containing polymers, the oxidation speed and products of dopa are not well regulated, leading to a mixture of various oxidation products[48,49].

In this work, we develop a novel electro-oxidation approach to controllably oxidize dopa to dopaquinone for the construction of hydrogel tapes that combine instant and robust adhesion with fault-tolerant operation. Compared to widely used chemical oxidization, electro-oxidation not only promotes the production

of dopaquinone but also reduces the generation of dopa oligomers, allowing the dopa groups to be efficiently utilized for interfacial crosslinking. The resulting hydrogel tapes form instant strong adhesion with various organic and inorganic surfaces through non-covalent interactions. The tapes can be repositioned multiple times without showing an obvious decrease in binding strength. After gradually forming covalent linkages, the hydrogel tapes show an increased adhesion strength of ~1268 J m$^{-2}$, outperforming many reported bio-adhesives[50–55]. We further demonstrate the feasibility of using the hydrogel tapes as tissue sealants and adhesives for soft and implantable devices both in vitro and in vivo. Finally, we show that the hydrogel tapes are biocompatible, degradable and suitable for various biomedical applications.

## Results and discussion

**Design and mechanism of the hydrogel tape**. The adhesive hydrogel tapes were made of bovine serum albumin (BSA), electro-oxidized alginate-dopa (see Supplementary Fig. 1 for synthesis details) and polyacrylic acid (PAA) (Fig. 1a). The hydrogels were covalently crosslinked between the amino groups on the BSA surface and the carboxyl groups of alginates and/or PAA in the presence of N-hydroxy succinimide (NHS) and 1-(3-dimethylaminopropyl)-3-ethylcarbodiimide hydrochloride (EDC). The entanglement of the polymers as well as the hydrogen bonding among dopa, amine, and carboxyl groups provided physical crosslinks (top panel of Fig. 1a). Moreover, PAA can absorb surface water, facilitating the formation of instant, dynamic, and strong interfacial bonds with dopa (middle panel of Fig. 1a)[30]. The alginate-dopa was partially electrically oxidized using a galvanic cell prior to hydrogel preparation (Supplementary Fig. 2). Electro-oxidation ensured that most of the products were dopaquinone, which was in sharp contrast to the chemical oxidation products (Supplementary Fig. 3a–e). The chemical oxidation of dopa by potassium periodate (KIO$_4$) mainly led to the formation of catechol dimers that cannot form covalent bonds with amino or thiol groups. However, electrochemical oxidation was slower than the chemical oxidation process. The electrochemical oxidation rate depends on many factors, including electrolyte resistance of the electrode, the surface adsorption of reactants, and the mass transfer of the reactants from bulk solution to the electrode surface. It is possible to improve the electrochemical oxidation speed in the future by optimizing the surface properties of the electrode. Nonetheless, the electro-oxidation method allowed us to selectively and controllably oxidize dopa to dopaquinone without producing other side products (Supplementary Fig. 3f–i). The electro-oxidation of dopa on electrodes has been studied by many researchers[56–58]. For electro-oxidation of dopa adsorbed on the electrode, dopa is first converted into dopaquinone. When dopaquinone on the electrode surface reaches a high concentration, polymerization/crosslinking reaction of dopa and dopaquinone takes place. The crosslinking reaction requires that two dopa/dopaquinone molecules are adsorbed in proximity on the electrode, which is difficult to achieve practically. Therefore, the dominant product is dopaquinone. In contrast, the chemical oxidation takes place in solution, in which dopa and dopaquinone diffuse freely, inevitably leading to the formation of dimer.

Dopaquinone on hydrogel surfaces can form covalent bonds with amino groups on tissue surfaces to establish long-term strong adhesion (bottom panel of Fig. 1a). The dopaquinone inside the hydrogels can form hydrogen bonds with unoxidized dopa and covalent bonds with the amino groups on BSA surfaces to further strengthen the hydrogel network (Fig. 1a). Due to the slow reaction of dopaquinone and amine or thiol on the tissue

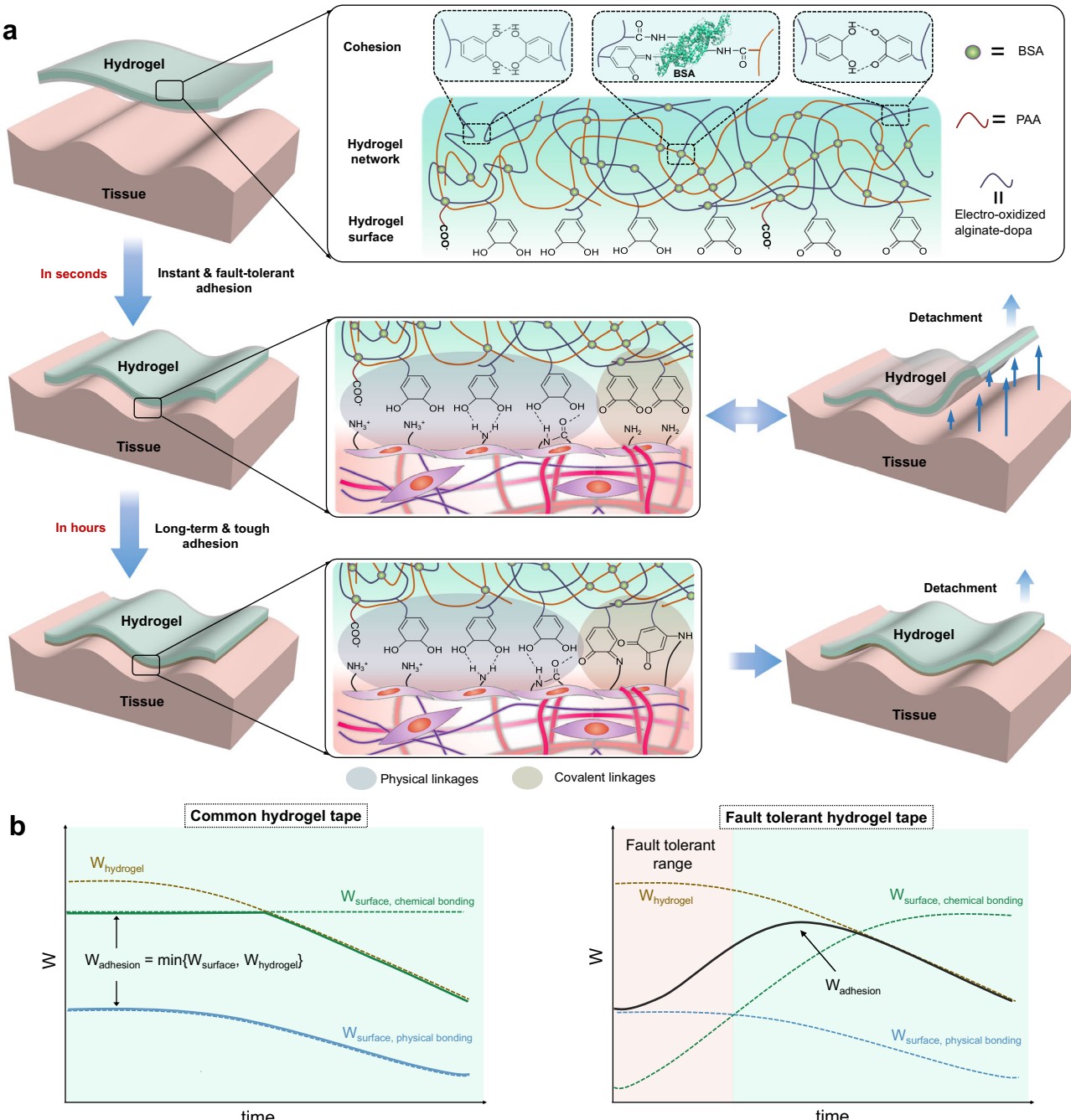

**Fig. 1 Schematic illustration of the adhesion and fault-tolerant mechanism of Electro-Ox hydrogel tape. a** Adhesion mechanism of Electro-Ox hydrogel tape. The hydrogel is mainly crosslinked by covalent bonds between BSA and electro-oxidized alginate-dopa or PAA formed via EDC-NHS. The conjugation between amine groups from BSA and dopaquinone generated during electro-oxidation further strengthens the hydrogel tapes. The unfolding of BSA and rupture of hydrogen bonds between dopa/dopaquinone can dissipate considerable energy and endow the hydrogel tape with high toughness. The adhesion between the hydrogel tape and tissue surface is time dependent. First, mainly non-covalent interactions (ionic interactions, cation–π interactions, and hydrogen bonds) are formed in seconds. As the reaction between amines from the tissue surface and dopaquinone in the hydrogel proceeds, the surface bonding is gradually enhanced in hours. **b** Comparison of the time-dependent adhesion strength of common hydrogel tapes using either physical bonding or chemical bonding for surface adhesion and the fault tolerant hydrogel tapes using a time-dependent formation of covalent bonding for surface adhesion. The overall adhesion strength ($W_{adhesion}$, solid line) depends on the lower one between the break strength of the hydrogel ($W_{hydrogel}$, dash line) and the surface bonding energy ($W_{surface}$, dash line). Physical surface bonding allows to detach the hydrogels freely but cannot ensure strong adhesion. In contrast, chemical surface bonding ensures strong adhesion but is not detachable. Due to the slow formation of chemical bonds with surface, the fault-tolerant hydrogel tapes can provide strong adhesion as instant chemical bonding hydrogel tapes yet have a time window for reversible detachment, thus combine the merits of both physical and chemical bonding-based hydrogel tapes.

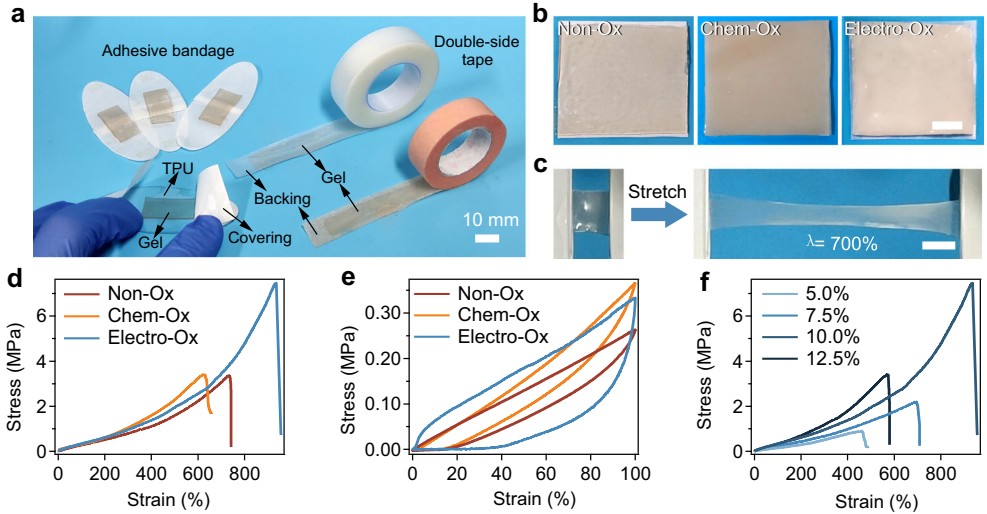

**Fig. 2 Mechanical properties of the adhesive hydrogel tapes based on electro-oxidized alginate-dopa. a** Adhesive bandages and double-side tapes made of Electro-Ox hydrogels. **b** Images of the Non-Ox, Chem-Ox, and Electro-Ox hydrogel tapes on white patches. Scale bar = 10 mm. **c** Images of the Electro-Ox hydrogel tape stretched to more than seven times the original length. Scale bar = 10 mm. **d** Typical tensile stress–strain curves of the Non-Ox, Electro-Ox and Chem-Ox hydrogel tapes at an alginate-dopa concentration of 10 w/v%. **e** Typical stretch–relaxation curves of Non-Ox, Chem-Ox, and Electro-Ox hydrogel tapes. **f** Typical tensile stress–strain curves of Electro-Ox hydrogel tapes at various mass concentrations of electro-oxidized alginate-dopa (5.0, 7.5, 10.0, and 12.5 w/v%).

surfaces, we expect the hydrogel tapes can provide long-term strong adhesion as chemical bonding hydrogel tapes yet have a time window for reversible detachment, thus combining the merits of both physical and chemical bonding-based hydrogel tapes (Fig. 1b). The hydrogels containing electro-oxidized alginate-dopa are named Electro-Ox hydrogels hereafter. For comparison, hydrogels containing unoxidized alginate-dopa or chemically oxidized alginate-dopa (named Non-Ox and Chem-Ox hydrogels, respectively) were also studied.

**Structure and physical properties**. We first characterized the structure and physical properties of the hydrogel tapes. Thanks to the excellent flexibility, the hydrogels can be fabricated into various forms such as bandage like strips and double-side tape like rolls (Fig. 2a). The color of the Chem-Ox and Electro-Ox hydrogels was darker than that of the Non-Ox hydrogels due to the oxidation of dopa (Fig. 2b). Scanning electron microscopic (SEM) images indicated that the pore sizes of the Electro-Ox and Chem-Ox hydrogels were smaller than those of the Non-Ox hydrogels, presumably due to their higher crosslinking density (Supplementary Fig. 4). The swelling ratios and solid contents of the three hydrogels were comparable (Supplementary Fig. 5). Due to the high crosslinking density of the Electro-Ox hydrogel tapes, their maximum swelling ratios were in the range of 2.5–3.2. This property greatly prevented the delamination of tape from wet tissues during long-term in vivo applications.

We then measured the mechanical properties of the hydrogel tapes. The Electro-Ox hydrogel tapes could be stretched to more than seven times the original length without causing any cracks (Fig. 2c), demonstrating their high stretchability. The fracture strain and fracture stress of the Electro-Ox hydrogel tapes were significantly higher than those of the Chem-Ox and Non-Ox hydrogel tapes (Fig. 2d and Supplementary Fig. 6a). The toughness of the Electro-Ox hydrogel reached 22.6 MJ m$^{-3}$ (Supplementary Fig. 6b). The stress–strain curves of all hydrogel tapes exhibited obvious hysteresis, suggesting that the tapes contained abundant non-covalent interactions that can be ruptured upon stretching to dissipate energy (Fig. 2e and Supplementary Fig. 6c). The Electro-Ox hydrogel tapes exhibited

the highest dissipated energy (0.13 MJ m$^{-3}$) and the largest dissipation ratio (72.6%). We further optimized the mechanical properties of the Electro-Ox hydrogel tapes by varying the amount of alginate-dopa (Fig. 2f) or the total solid content (Supplementary Fig. 7). The Young's modulus increased with increasing alginate-dopa or total mass concentrations (Supplementary Figs. 6d and 7b). However, the maximum strain was negatively correlated with the total mass concentration[59,60]. Therefore, the highest fracture energy was achieved at an alginate-dopa concentration of 10% (w/v) and a total solid content of 50% (w/v) (Supplementary Figs. 6 and 7). The high mechanical strength and toughness as well as the significant energy dissipation make the hydrogel tapes suitable for strong adhesion[61].

**Adhesion performance**. Next, the adhesion performance of the Electro-Ox hydrogel tapes was evaluated at different timescales. We used wet porcine skin as the model tissue because its mechanical and biological properties are similar to those of human skin[62]. We first allowed two pieces of porcine skin to adhere using Electro-Ox hydrogel tape for <10 s and measured the adhesion strength under cyclic attachment/detachment at a short timescale (Fig. 3a). The adhesion strength was ~28 kPa for the first cycle, decreased slightly to 90% after 6 cycles, and decreased to ~66% after 20 cycles (Fig. 3b). We also studied whether the adhesion strength in the second cycle was affected by the time that the adherends remained formed in the previous cycle before being detached (Fig. 3c). The adhesion strength in the second cycle retained >70% of the initial value when the tape was adhered for 20 min and was still ~50% when the tape was adhered for 1 h. These results confirmed that the Electro-Ox hydrogel tapes have a strong instant binding strength and are detachable and reusable.

We then used three different types of mechanical tests (peel test for interfacial toughness, lap shear test for shear strength, and tensile test for tensile strength) to evaluate the long-term adhesion performance of the hydrogel tapes after being adhered for 24 h (Fig. 3d–f and Supplementary Fig. 8)[29]. As summarized in Fig. 3f, the shear and tensile strength between the Electro-Ox

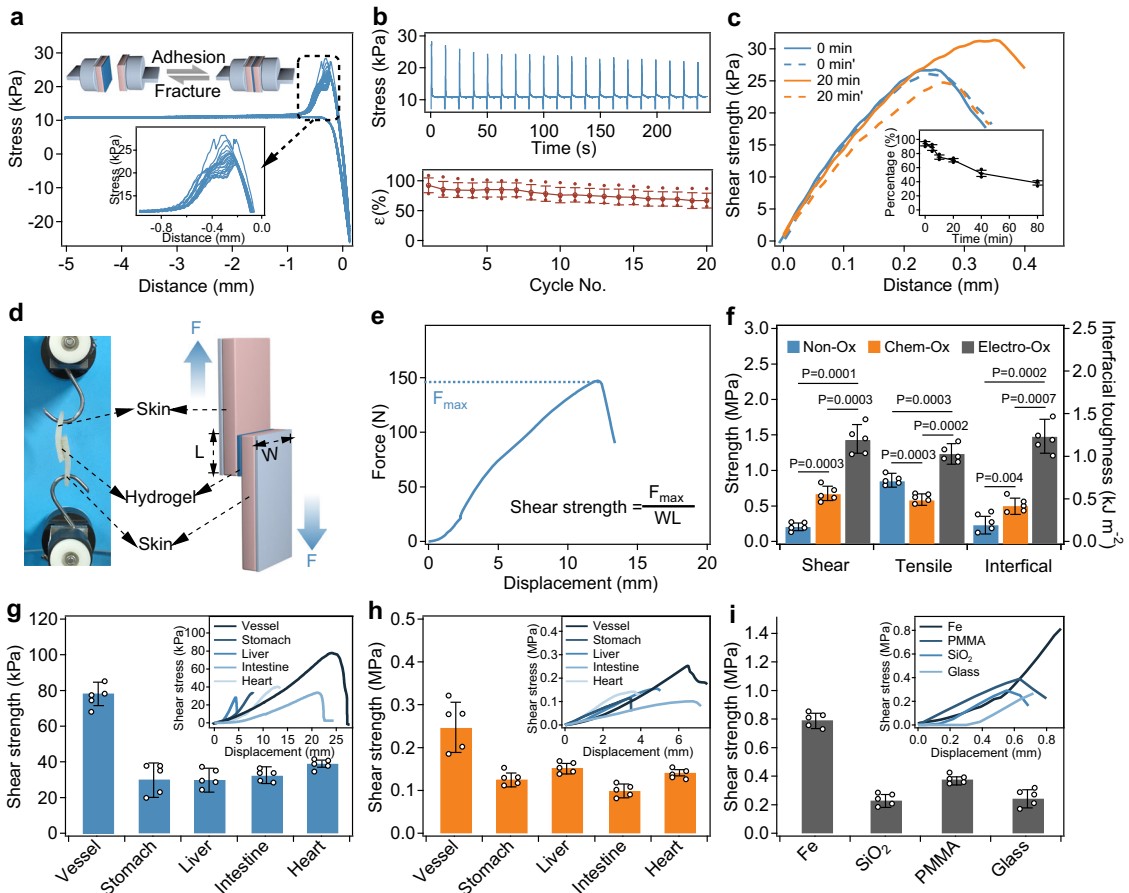

**Fig. 3 Adhesion performance of the Electro-Ox hydrogel tape. a** Cyclic compression and fracture curves versus displacement of instant adhesion for porcine skin using Electro-Ox hydrogel tape in 20 cycles. The top inset corresponds to a schematic of the compression and fracture cycles. The bottom inset corresponds to the magnified fracture curves. **b** Cyclic compression and fracture curves versus time (top) and normalized tensile strength ($\varepsilon$, bottom) of instant adhesion for porcine skin using Electro-Ox hydrogel tape in 20 cycles. Values represent the mean and standard deviation ($n = 3$ independent samples). **c** Typical lap shear curves of instant adhesion (0 min), adhesion after the fracture of initial adhesion (0 min'), adhesion after curing for 20 min (20 min), and instant adhesion after the fracture of 20 min adhesion (20 min'). The inset corresponds to the normalized tensile strength of a second instant adhesion event after different times (normalized 0 min', 5 min', 10 min', 20 min', 40 min', and 80 min'). Values represent the mean and standard deviation ($n = 3$ independent samples). **d** Image and schematic for the measurement of shear strength based on the standard lap shear test. F force, W width, L length. **e** Typical force–displacement curve recorded in the lap shear test and the determination of shear strength. **f** Shear strength, tensile strength, and interfacial toughness of long-term adhesion for different hydrogels. P values, two-tailed Student's t test. No adjustments were made for multiple comparisons. **g, h** Shear strength of short-term (**g**) and long-term (**h**) adhesion for different porcine organs using Electro-Ox hydrogel tape. **i** Shear strength of long-term adhesion for different substrates (Fe, SiO$_2$, PMMA, and glass) using Electro-Ox hydrogel tape. Insets of **g–i** correspond to the typical force–displacement curves recorded in the lap shear tests. Values in **f–i** represent the mean and standard deviation ($n = 5$ independent samples).

hydrogel tape and the wet porcine skin were ~1.46 and 1.25 MPa, which are comparable with that of commonly used sutures (Supplementary Fig. 9, ~2.0 MPa). The interfacial toughness of the Electro-Ox hydrogel tape and the porcine skin reached 1268 J m$^{-2}$, outperforming many reported bio-adhesives[48,63–68]. This high shear strength, tensile strength, and interfacial toughness indicated that the Electro-Ox hydrogel tape can establish strong and tough adhesion between wet porcine skin samples. In contrast, the shear strength, tensile strength, and interfacial toughness of the Chem-Ox (shear strength: 0.68 MPa, tensile strength: 0.62 MPa, interfacial toughness: 0.45 kJ m$^{-2}$) and Non-Ox hydrogel tapes (shear strength: 0.24 MPa, tensile strength: 0.87 MPa, interfacial toughness: 0.23 kJ m$^{-2}$) were significantly smaller due to the presence of less dopaquinone than that in the Electro-Ox hydrogel tape. Moreover, the adhesion strength could be further optimized by varying the composition and electro-oxidation time (Supplementary Fig. 10). The adhesion strength of Electro-Ox tapes first increased with the increase of the oxidation time in 6 h and then decreased gradually

afterwards. We inferred that the dopaquinone groups on alginate reached the maximum at the oxidation time of 6 h. Meanwhile, still quite some dopa remained unoxidized and they contributed to the short-term adhesion. However, further extending the electro-oxidation time would lead to the formation of intramolecular crosslinking in alginate-dopa, thus reducing available dopaquinone and dopa groups for surface bonding and drastically decreased the adhesion strength. The proposed mechanism was supported by the high-performance liquid chromatographic traces of electro-oxidized N-acetyldopamine products (Supplementary Fig. 3b–e) at different electro-oxidation time. The peak corresponding to quinone decreased while that belonging to dimer increased, indicating the conversion of dopaquinone and dimer.

The excellent adhesion strength mainly attributed to the strong adhesion and cohesion of the hydrogels. The outstanding shear and tensile strength of the Electro-Ox hydrogel comes from the unique crosslinking mechanism and the network structures. In the Electro-Ox hydrogel, the covalent crosslinks include the

amide bond between the amino groups on the BSA surface and the carboxyl groups of alginates and/or PAA and some dopaquinone–amine adducts. The physical crosslinks include the hydrogen bonding between dopa and dopaquinone[42], π–π stacking among dopa groups[39], the charge–charge interaction between BSA and PAA/alginate, and numerous physical interactions in the folded BSA structure. The rupture of the physical interactions prior to the break of covalent bonds can efficiently dissipate energy. Moreover, BSA is 7.1 nm in diameter and serve as a crosslinking hub. Similar to the nanoparticle-crosslinked hydrogels[69], the nanosized BSA can prevent crack propagation and further increase the tensile strength. Moreover, the formation of abundant dopaquinone and amine/thiol covalent bonds ensures long-term strong surface bonding. The combined strong surface bonding and the strength of the hydrogel matrix ensure the overall shear and tensile strength of the hydrogel tapes over megapascal. Even though the Chem-Ox and Non-Ox hydrogels also exhibited considerable adhesion strength, they are limited for certain applications. For non-Ox hydrogels, the adhesion strength to organic tissues is relatively weak compared to Chem-Ox and Electro-Ox hydrogels as shown in Fig. 3f. For Chem-Ox hydrogels, the uncontrollable oxidation of dopa (Supplementary Fig. 3) and the generation of dimer would affect the adhesion performance on inorganic substrates compared to Non-Ox and Electro-Ox hydrogels. It is worth mentioning that Electro-Ox hydrogels can quickly absorb the interfacial water to break the interfacial hydration layer, leading to the direct contact of the adhesive groups with the substrates (Supplementary Fig. 11). Besides, the adhesion strength of Electro-Ox hydrogels tapes on organic tissues might also be affected by the ambient pH and enzymes (Supplementary Fig. 12). The shear strength of the hydrogel tape-adhered porcine skins reduced to ~600 kPa at pH 1.0. The adhesion strength of porcine skin covered with trypsin solution decreased markedly to 152 kPa. The reduced adhesion strength can be explained by the enzymatic digestion of BSA by trypsin. Similarly, the adhesion strength of porcine skin covered with the raw bile extract remained at ~395 kPa. Although bile does not contain enzymes that can digest the hydrogels, the protein components in bile can still partially block the formation of covalent linkage between the hydrogel tape and the tissue surfaces. Note that even the adhesion strength was much reduced at low pH or in the presence of enzymes, it was still comparable with or even higher than the adhesion strength of some commercial bio-adhesives such as polyethylene-glycol and fibrinogen-based tissue adhesives[29].

We then characterized the mechanical stability of long-term tissue adhesion by the Electro-Ox hydrogel tape under a dynamic mechanical load. Two pieces of wet porcine skin joined by hydrogel tape were cyclically stretched to 20% strain and then relaxed (Supplementary Fig. 13a). The shear strength slightly decreased with increasing cycle number and retained >85% of the original value even after 5000 cycles, further demonstrating the long-term stability of the hydrogel tapes in a dynamic mechanical environment (Supplementary Fig. 13b, c). It is worth noting that the detachment of the Electro-Ox hydrogel tapes adhered on the organic tissues can be achieved by adding dithiothreitol (DTT) solutions to the interface. The bonds formed between dopaquinone and amine or thiol on tissue surfaces are dynamic covalent bonds. They can be disrupted by the competitive binding of DTT. As shown in Supplementary Fig. 14, the shear strength of the hydrogel tape-adhered porcine skins decreased by >97% (from 1.45 to 0.032 MPa) after being soaked in a phosphate buffer solution (PBS) solution containing 1 mM DTT for 1 h. This makes it easy to remove the adhered Electro-Ox hydrogel tape and causes ignorable tissue injury. As shown in the SEM images of the porcine skin surfaces after removing the hydrogel tape, no obvious injury can be observed (Supplementary Fig. 15).

We then evaluated the short- and long-term shear strength between the Electro-Ox hydrogel tapes and various ex vivo tissues, including porcine vessels, stomach, liver, intestine, and heart. As shown in Fig. 3g, h, the shear strength for short-term adhesion (<10 s) was >30 kPa for all tissues (~78 kPa for vessels, ~31 kPa for the stomach, ~32 kPa for the liver, ~33 kPa for the intestine, and ~40 kPa for the heart), and the shear strength for long-term adhesion (24 h) was >100 kPa (~250 kPa for vessels, ~130 kPa for the stomach, ~155 kPa for the liver, ~101 kPa for the intestine, and ~144 kPa for the heart). The shear strengths of the Electro-Ox hydrogel tapes for different tissues were significantly higher than those of fibrinogen-based bio-adhesives (Supplementary Fig. 16a, c), which are usually <30 kPa (~12.6 kPa for vessel, ~19.6 kPa for stomach, ~28.6 kPa for liver, ~6.3 kPa for intestine, and ~8.8 kPa for heart). Besides, the adhesion strength of Electro-Ox hydrogel tapes was not significantly affected by the blood on tissue surfaces (Supplementary Fig. 17). Moreover, serum proteins that had been adsorbed into the hydrogels can react with dopaquinone to provide additional crosslinks and reduce the swelling ratio of the hydrogel tape (Supplementary Fig. 18). These features make Electro-Ox hydrogels ideal candidates for hemostatic dressings. The short-term adhesion of the Electro-Ox hydrogel tapes on different tissues were >30 kPa, which is strong enough for instant tissue adhesions. The initial strong adhesion is based on physical interactions. These interactions ensure reversible attachment and detachment of the hydrogel tapes. However, these interactions are severely weakened due to swelling of the adhesives or bleeding from the tissue[24–28]. Therefore, slow formation of strong covalent bonds is critical for the potential applications, such as wound closure, fixing implantable devices, and hemostasis. The formation of covalent bonding between dopaquinone and amino/thiol increased the adhesion strength to >100 kPa, making the hydrogel tapes suitable for long-term applications.

The Electro-Ox hydrogel can also establish stable adhesion between wet tissues and other solid surfaces that are widely found in various electronic devices (Fig. 3i). The shear strength of the long-term adhesion reached 803, 243, 390, and 249 kPa for iron (Fe), silicon dioxide ($SiO_2$), polymethyl methacrylate (PMMA), and glass substrates, respectively. Even though the shear strength of the Electro-Ox hydrogel tapes on these solid surfaces were smaller than those of cyanoacrylate-based glue (Supplementary Fig. 16b, d; ~2.9 MPa for Fe, ~3.6 MPa for $SiO_2$, ~8.0 MPa for PMMA, and ~3.6 MPa for glass), the adhesion strength is strong enough for fixing electronic devices[35].

**Potential applications**. We next evaluated the potential applications of the Electro-Ox hydrogel tape for tissue adhesion using ex vivo porcine tissue models. The Electro-Ox hydrogel tape can be applied to seal a water-leaking stomach (Fig. 4a and Supplementary Movie 1) and air-leaking lung (Fig. 4c and Supplementary Movie 2) in 30 s. Moreover, the sealed stomach and lung can perfectly hold water or air without any leakage after sealing, as monitored using the changes in water level and volume (Fig. 4b, d). Moreover, the peak pressure of sealed lung can remain stable under cyclical insufflations lasting for 8 h (Frequency: 13 breathe per minute; tidal volume ($V_t$): 600 mL; maximum pressure ($p_{Max}$): 40 mBar; positive end expiratory pressure: 5 mBar), suggesting the robustness of the lung sealing (Supplementary Fig. 19). The observed burst pressures for short-term sealing of stomach and lung were ~20.6 and ~17.9 kPa, respectively (Supplementary Fig. 20). These values then increased to ~69.0 and ~58.0 kPa, respectively, for long-term sealing. Electro-Ox hydrogel tape can also be used to adhere devices onto tissue surfaces to monitor dynamic motion. As an example, we adhered

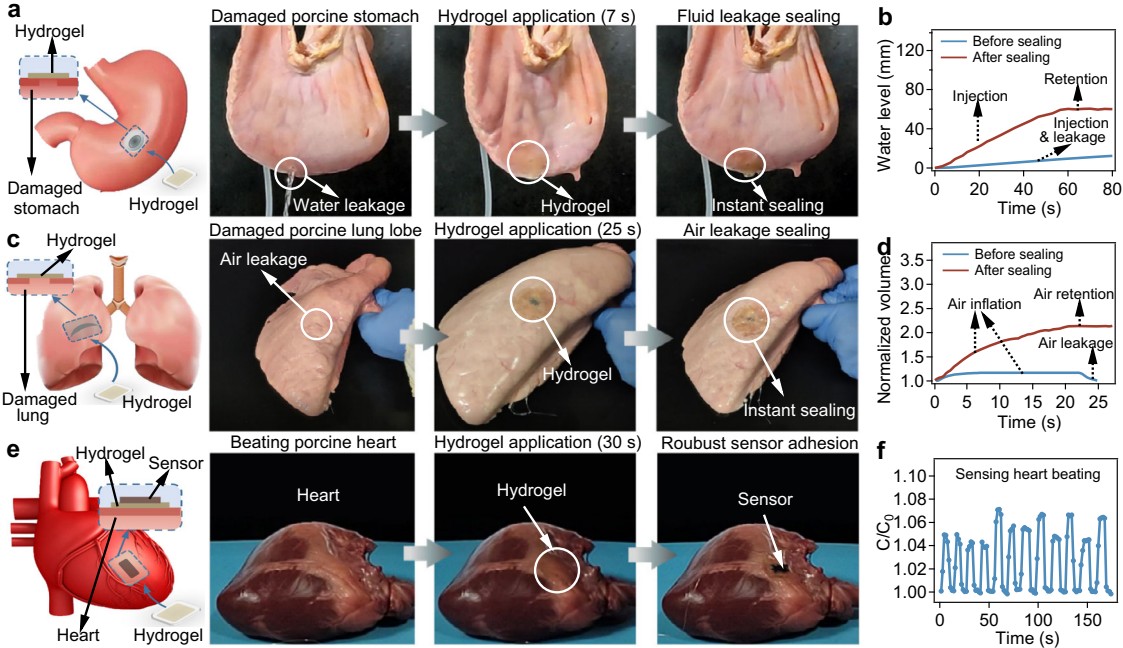

**Fig. 4 Applications of the Electro-Ox hydrogel tape to tissue adhesion. a** Sealing of a water-leaking porcine stomach using Electro-Ox hydrogel tape.
**b** Water level in the stomach corresponding to the sealing process in **a**. **c** Sealing of an air-leaking porcine lung using Electro-Ox dopa hydrogel tape.
**d** Volume change in the lung corresponding to the sealing process in **c**. **e** Electro-Ox hydrogel tape-mediated adhesion of a hydrogel-based strain sensor on a beating porcine heart driven by an air pump. **f** Normalized capacitance ($C/C_0$) of the strain sensor adhered to the porcine heart over time corresponding to mimicked heart beating in **e**.

a capacitive strain sensor on a porcine heart via Electro-Ox hydrogel tape (Fig. 4e). The heart was cyclically pressurized by intermittent airflow to mimic heartbeats. The strain sensor attached to the beating heart surface could monitor the motion of the heart through the synchronous deformation of the pericardium, the Electro-Ox hydrogel tape, and the flexible sensor (Fig. 4f and Supplementary Movie 3).

We further evaluated whether the hydrogel tapes were suitable for in vivo adhesion and hemostasis. After being pressed for 7 s, the Electro-Ox hydrogel tape stably adhered to a rabbit gastrocnemius surface (Fig. 5a). After implantation for 3 days in vivo, the hydrogel tape reached swelling equilibrium and remained adhered on the surface (Fig. 5b and Supplementary Movie 4). When the hydrogel tape was peeled off, the fracture appeared inside the hydrogel layer instead of at the surface. We further adhered a metal sheet (weight of ~0.52 g and diameter of ~9 mm) to the epicardial surface of a beating rabbit heart using Electro-Ox hydrogel tape to mimic device adhesion in vivo (Fig. 5c). The metal sheet stably adhered to the beating heart after gentle pressing for 7 s (Fig. 5d and Supplementary Movie 5). After beating for 3 h (~36,000 beats), the metal sheet still adhered at the same position. After the adhesion for 72 h, the shear strength was ~80 kPa as indicated by the lap shear test (Supplementary Fig. 21). However, even the hydrogel tapes were removed from the gastrocnemius or heart at this stage, no obvious injuries were observed, suggesting that the covalent bonding of hydrogel tapes to the tissue surfaces did not cause deep tissue injury (Supplementary Fig. 22a, b). These results demonstrate the suitability of applying this hydrogel tape for many challenging in vivo applications. Furthermore, we evaluated the hemostasis of the bleeding carotid artery of a living pig using Electro-Ox hydrogel tape (Fig. 5e). After gently compressing the Electro-Ox hydrogel tape on a bleeding carotid artery, the bleeding was stopped without any leakage and no rebleeding was observed in the following 15 min (Fig. 5f, Supplementary Fig. 23a, and

Supplementary Movie 6). Comparing to the hemostasis using manual pressure, the amount of bleeding using the hydrogel tape decreased by >90% (Supplementary Fig. 23b, c). The histologic response of the artery to the hydrogel tape was also investigated (Supplementary Fig. 22c). No obvious tissue damage or pathological change was observed after applying the hydrogel tape for 72 h, indicating that the hydrogel tapes are suitable for hemostasis. These results demonstrated that the hydrogel combines great mechanical strength, rapid surface bonding, and long-term mechanical stability and is suitable for applications in dynamic and stress-bearing tissues.

**Biocompatibility and biodegradability.** Finally, we investigated the biocompatibility and biodegradability of the Electro-Ox hydrogel tape in vitro and in vivo (Fig. 6). For the in vitro biocompatibility tests, mouse embryo osteoblast (MC 3T3) cells and mouse embryonic fibroblast cells were cultured on Electro-Ox hydrogels for 24 h before the cell viabilities were determined using live/dead cell staining (Fig. 6a). Nearly no dead cells could be observed on the hydrogel tapes for both cells, and the cell viability was >95% compared to the control group (Fig. 6b and Supplementary Fig. 24). Moreover, the Electro-Ox hydrogel showed protease-dependent degradability. The Electro-Ox hydrogel tape remained stable when placed in PBS or simulated body fluid (SBF) buffer at 37 °C for 10 days (Fig. 6c). Upon the addition of collagenase, the hydrogel tape quickly lost 40% of its weight in 10 days. Because PAA and its derivatives are not degradable under physiological conditions[70] and the degradation of alginate in vivo is slow and uncontrollable[71], we think the degradation of the hydrogels is mainly due to the enzymatic digestion of BSA, which leads to the decomposition of the hydrogel network.

We finally evaluated the in vivo biocompatibility and biodegradability of the tape based on dorsal subcutaneous implantation in a rat model (Fig. 6d–s and Supplementary

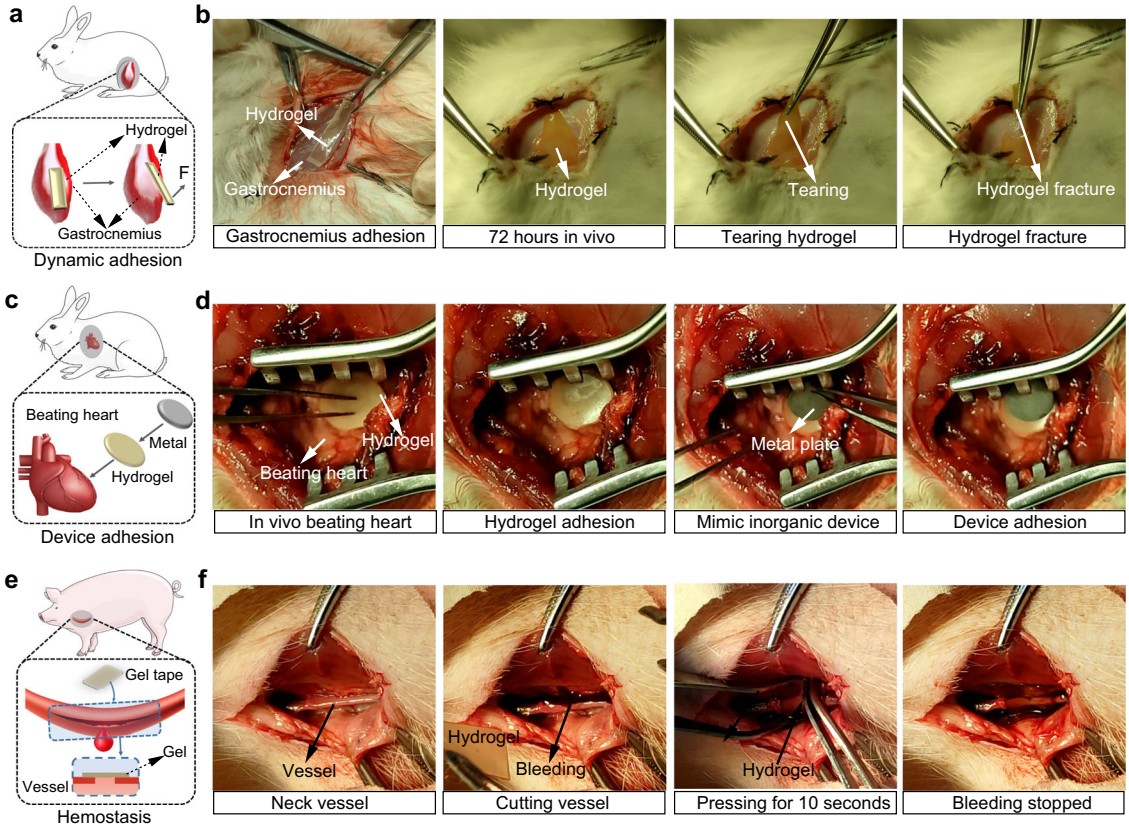

**Fig. 5 In vivo adhesion and hemostasis of Electro-Ox hydrogel tape. a** Schematic illustration of adhesion on rabbit gastrocnemius. **b** Adhesion of Electro-Ox hydrogel tape on rabbit gastrocnemius before and after 72 h in vivo. **c** Schematic illustration of device adhesion on a beating rabbit heart. **d** Adhesion of a metal sheet on a beating rabbit heart via Electro-Ox hydrogel tape in vivo. **e** Schematic illustration of hemostasis on the bleeding carotid artery of a pig. **f** Hemostasis of the bleeding carotid artery of a pig using Electro-Ox hydrogel tape in vivo.

Fig. 25). The groups of the sham surgery and commercial tissue adhesives (fibrinogen- and cyanoacrylate-based bio-adhesives) as well as the groups without surgery were also investigated for comparison. As indicated by the histological assessment (hematoxylin and erosion (H&E) staining) results, the hydrogel tapes did not cause any major damage to the surrounding dermal and muscular layers (Fig. 6d). A mild chronic inflammatory response could be found in all the groups on the first day, as indicated by the presence of lymphocytes cells and moderate eosinophilic response (Fig. 6d–g). No necrosis of the skeletal muscle and skin was observed. The inflammatory response was considerably weakened with increasing time and almost disappeared after 14 days. The immunofluorescence staining of tumor necrosis factor (TNF)-α and interleukin (IL)-1β staining for the samples in 1 day after subcutaneous implantation in the backs of Sprague Dawley rats were also presented (Supplementary Figs. 25 and 26). A low expression level of TNF-α and 1L-1β around the hydrogel tapes were observed similar to those of the commercially available bio-adhesives and sham surgery. Moreover, the Electro-Ox hydrogel tapes exhibited fast degradation in 14 days, similar to that of the fibrinogen-based bio-adhesives (Fig. 6d, e). Formation of collagenous capsules and granulation tissue around the implantation sites was observed after 14 days, indicating the breakdown of the hydrogel tapes and the healing of the damaged tissues. In contrast, the cyanoacrylate-based bio-adhesive did not degrade in 14 days after implantation (Fig. 6f).

Furthermore, we conducted systemic toxicity tests to evaluate the generalized biological effects of the hydrogel tape. The blood biochemistry (alanine transferase, aspartate transferase, urea, creatinine, $Ca^{2+}$ and $K^+$), cell (white blood cell count, red blood cell count, hemoglobin, and lymphocyte), and coagulation (activated partial thromboplastin time and fibrinogen) indicators did not show obvious change after implantation (Fig. 6h–s). The similar values of the indicators as those of the commercially available bio-adhesives and sham operation demonstrated that the implantation of the hydrogel tape did not affect the function of heart, liver, and kidney, as well as the coagulation of blood.

At last, we further evaluated the histologic response of the major organs (heart, liver, spleen, lung, kidney, stomach, and artery) to the chronic administration of the Electro-Ox hydrogel and other bio-adhesives (Supplementary Figs. 27 and 28). No obvious histologic response or pathological change to the major organs (heart, liver, spleen, lung, kidney, stomach, and artery) was observed in 14 days after implantation. Currently, we did not study the immune/healing response of the target tissues (stomach, lung, and artery) in direct contact to the hydrogel tapes. Although the ex vivo experiments show the possible applications of the Electro-Ox hydrogel tapes on these organs, we realize that the long-term biocompatibility and host immune/healing response should be more rigorously investigated before the clinic applications of the hydrogel tapes.

In this work, we report mechanically robust hydrogel tapes suitable for rapid and strong adhesion to various wet tissues and solid electronic devices. In contrast to other hydrogel tapes that use fast surface covalent bonding (e.g., EDC/NHS-catalyzed amide bond formation), we introduced a slow covalent bonding reaction between dopaquinone and the amino groups of wet tissue surfaces. Due to the moderate reaction rate, the tapes can be repositioned multiple times without showing an obvious decrease in binding strength relative to the initial value.

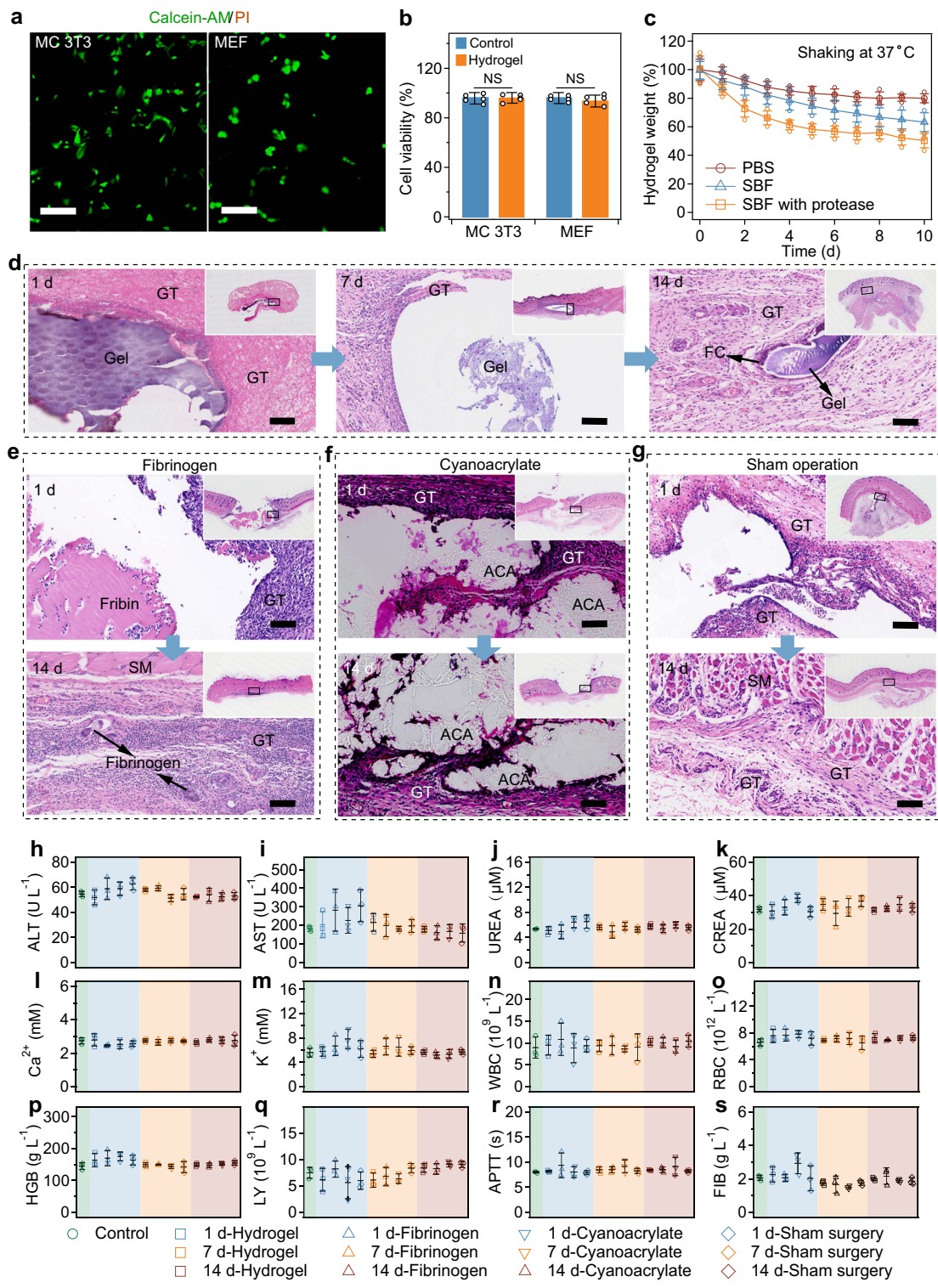

Moreover, the hydrogel tapes showed a great adhesion strength of ~1268 J m$^{-2}$ after covalent linkages were formed in hours. The great blood adsorbing ability of the hydrogel tapes allows them to function properly on blood-covered surfaces. We further demonstrated the successful use of the hydrogel tapes as tissue sealants and adhesives for wearable and implantable devices, even on dynamic and bleeding tissue surfaces. The hydrogel tapes were biocompatible, degradable, and suitable for various biomedical applications. The demonstration of the combination of instant and strong wet adhesion with fault-tolerant practical surgical operation may inspire the design of next-generation hydrogel tapes that can function in challenging surgical conditions.

## Methods

**Preparation of hydrogel.** In a typical preparation of the hydrogel, BSA, PAA, and dopa-containing polymer (electro-oxidized, chemically oxidized, or unoxidized alginate-dopa) were dissolved in ddH$_2$O to concentrations of 20, 60, and 20 mg mL$^{-1}$, respectively. Then 25 mL of the mixture was dried in a petri dish (diameter of 60 mm) for 24 h at 3% humidity and 25 °C to obtain dry tape samples. Then PBS solution (5 mL) containing EDC (100 mM) and NHS (100 mM) was applied to the tape to crosslink the BSA, PAA, and alginate. The hydrogel tape was transferred

**Fig. 6 Biocompatibility and biodegradability of Electro-Ox hydrogel tapes in vitro and in vivo. a, b** Fluorescence microscopic images (**a**) and cell viability (**b**) of MC 3T3 and MEF cells cultured on Electro-Ox hydrogels. The living and dead cells were stained with a live/dead assay (Calcein-AM/PI Double Staining Kit) after 24 h of culture. P values, two-tailed Student's t test. No adjustments were made for multiple comparisons. Scale bar = 100 μm. **c** In vitro biodegradation of Electro-Ox hydrogel tape in phosphate buffer saline (PBS) and simulated body fluid (SBF) with or without protease. Values in **b**, **c** represent the mean and standard deviation (n = 3–5 independent experiments). **d** Representative histological images stained with hematoxylin and eosin (H&E) for assessment of the biocompatibility and biodegradation of the Electro-Ox hydrogel tape in vivo at Days 1, 7, and 14 after subcutaneous implantation. GT and FC indicate granulation tissue and fibrous capsule, respectively. **e–g** Representative histological images of the control groups, **e** the fibrin-based glue, **f** the cyanoacrylate (ACA) glue, and **g** the sham surgery. SM and GT indicate skeletal muscle and granulation tissue, respectively. All histological experiments were repeated at least three times with similar results. Scale bar = 100 μm. **h–m** Blood biochemistry examination of the variation in ALT (alanine transferase), AST (aspartate transferase), UREA (urea), CREA (creatinine), $Ca^{2+}$, and $K^+$. **n–q** Hematological examination of the variation in WBC (white blood cell count), RBC (red blood cell count), HGB (hemoglobin), and LY (lymphocyte). **r, s** Coagulation action test of the variation in APTT (activated partial thromboplastin time) and FIB (fibrinogen). The blood of rats without any surgery was set as the control. In **h–s**, the samples were taken at Days 1, 7, and 14 after subcutaneous implantation and reported values represent the mean and standard deviation (n = 3 independent animals).

into $ddH_2O$ to achieve swelling equilibrium for 24 h, during which time the $ddH_2O$ was refreshed >6 times to remove the unreacted reactants. Finally, the obtained hydrogels were stored in a mixture of alcohol (75%) and $ddH_2O$ (25%) at room temperature before the experiments.

**Tensile test**. Tensile stress–strain measurements of the hydrogels were performed using a tensile-compressive tester (Instron-5944 with a 10 N sensor) in air at room temperature. Unless otherwise stated, the strain rate of stretching was maintained at ~10 mm $min^{-1}$ (1.0 mm $mm^{-1}$). The strain, $\lambda$, was defined by the distance displacement between the two clamps when the gel was deformed divided by the distance when the gel was undeformed. The Young's modulus corresponded to the approximate linear fitting value at a strain of 20%. The toughness was calculated from the area below the tensile stress–strain curve until fracture. The raw data of the mechanical tests was dealt with Microsoft Excel 2019 and the curves were drawn using Igor 6.37.

**Adhesion measurements**. Tissue samples were covered with PBS and stored in plastic bags at 4 °C before measurement. All the samples were prepared with the hydrogel sandwiched between the tissues and pressed for 30 s at a pressure of approximately 1.5 kPa. Then adhesion measurements were performed after 24 h to allow equilibrium swelling of the hydrogels in a wet environment. All adhesion measurements were performed using a mechanical testing machine (2 kN load cell, Instron 5944, US). The fat under the porcine skins was removed with blade and the dermal surface of the porcine skin is cleaned with alcohol and gauze before the experiments.

For the shear strength measurements, the width and length of the adhesion areas were approximately 8 and 10 mm, respectively. All experiments were conducted in air at 25 °C with a constant stretching rate of 50 mm $min^{-1}$ (Fig. 3d, e). The shear strength was determined as the maximum stress during the lap shear progress. For the lap shear experiments in the presence of DTT solutions, the porcine skin adhered using the hydrogel tapes was soaked in a PBS solution containing 1 mM DTT for 1 h before the shear strength was determined. For the lap shear experiments in the presence of trypsin, bile, or at low pH, the solution of trypsin (2.5 mg $mL^{-1}$ in PBS), the raw porcine bile extract, or a solution at pH 1.0 were applied to the porcine skin surface before the adhesion. For the interfacial toughness measurements, the samples were adhered with a width of 10 mm and tested by standard 180-degree peel tests (Supplementary Fig. 8a, b). All experiments were conducted in air at 25 °C with a constant stretching rate of 50 mm $min^{-1}$. The interfacial toughness was determined as two times the plateau stress. For the tensile strength measurements, the width and length of the adhesion area were approximately 8 and 8 mm, respectively. All experiments were conducted in air at 25 °C with a constant stretching rate of 50 mm $min^{-1}$ (Supplementary Fig. 8c, d). The tensile strength was determined as the maximum stress during the tensile stretching process.

For the cyclic stretch–relaxation test, two porcine skin tissues were adhered using the Electro-Ox hydrogel with an adhesion area of width 15 mm and length 30 mm. Then the adhered samples were cyclically stretched to a strain of 20% with an initial hydrogel length of 100% and released to the original length. After different numbers of loading cycles, the shear strength of the samples was determined using the shear strength measurements and normalized to the initial shear strength. During the tests, PBS was sprayed onto the samples using a humidifier to avoid dehydration.

**In vitro and in vivo biocompatibility and biodegradation**. In a typical biodegradation test in vitro, the hydrogel was cut into pieces with the same volume (0.5 mL), and the original dry weight of each hydrogel after washing with $ddH_2O$ was defined as $W_0$. Then a piece of Electro-Ox hydrogel (0.5 mL) was soaked in 50 mL of PBS (10 mM, pH = 7.4), SBF, or SBF containing 0.5 mM collagenase. The sample was incubated at 37 °C with a shaking rate of 220 r.p.m. At a certain time (t), the hydrogel was taken from the incubation medium and washed with $ddH_2O$

three times. The sample was lyophilized, and the weight was defined as $W_t$. The weight loss at time $t$ was determined as $\varepsilon = \frac{W_0 - W_t}{W_0} \times 100\%$.

For in vivo biocompatibility, hydrogel tapes and fibrinogen-based bio-adhesives with the width of 8 mm and length of 8 mm were prepared and stored in a mixture of alcohol (75%) and $ddH_2O$ (25%) prior to implantation. Then male Sprague Dawley rats aged 8 weeks (250 ± 10 g) were randomly divided into four groups (n = 15). (group A: the Electro-Ox hydrogel was administered; group B: the fibrinogen-based bio-adhesive was administered; group C: the cyanoacrylate-based bio-adhesive was administered; and group D: the sham surgery was performed). The rats were anesthetized with isoflurane. The back hair of the rats was removed, and each sample was inserted between the skin and muscle through a 1.5-cm skin incision at the center of the rat back. Then the incision was closed using interrupted sutures (3-0 Vicryl Plus, JNJ). After 1, 7, or 14 days, 5 of the rats were sacrificed, and target subcutaneous regions as well as major organs were obtained and fixed in 10% formalin for 24 h before histological analysis (H&E staining) and immunofluorescent staining (TNF-α and IL-1β). Blood samples were collected at the same time points for the systemic toxicity tests.

**In vitro tissue adhesion and hemostasis**. To evaluate the application of the Electro-Ox hydrogel in tissue adhesion and hemostasis, a series of in vitro experiments were performed. The freshly obtained organs were washed with PBS (10 mM, pH = 7.4) and patted down gently before applying the hydrogel tapes in the in vitro test. To prevent the dehydration of the organs, PBS (10 mM, pH = 7.4) was sprayed onto the organs frequently to keep the surface of the organs wet. For the sealing of a damaged stomach, a hole with a diameter of 10 mm was made in a porcine stomach. A bottle of water was poured into the stomach to show water leakage. Then Electro-Ox hydrogel tape (diameter: 30 mm) was applied to the hole and waited for 25 s. Upon pouring water into the stomach again, the leakage stopped. The water level inside was also monitored using a communicating pipe.

For the sealing of a damaged lung, a hole with a diameter of 15 mm was made in a porcine lung. Air was pumped into the lung through an air source (SPB-2, BCHP, China), and the volume of the lung barely increased due to air leakage. Then Electro-Ox hydrogel tape (diameter: 30 mm) was applied to the hole and waited for 30 s. When air was pumped into the lung again, air leakage stopped. The volume of the lung was monitored using image analysis with the ImageJ (V 1.8.0) software.

For the adhesion of a flexible strain sensor[72], hydrogel tape was applied to the surface of a heart, and a strain sensor was adhered to the hydrogel tape patch. Then the sensor was connected to an LCR meter (HIOKI-IM3536, Japan), and the heart was made to beat under intermittent airflow. The deformation of the heart was monitored via the capacitance change in the strain sensor.

**In vivo adhesion tests**. For in vivo adhesion on the gastrocnemius, hydrogels with a width of 5 mm and length of 10 mm were prepared prior to implantation. Then rabbits (9-week-old female New Zealand rabbit with a body weight of ~2.5 kg, n = 3) were anesthetized with isoflurane. The hair was removed before the operation. The skin of the gastrocnemius was cut, and hydrogel tape adhesion was achieved through a 2.5 cm incision. Then the wound was stitched using interrupted sutures (3-0 Vicryl Plus, JNJ). After 3 days, the wound was opened in the same region, and images were taken.

For in vivo adhesion on a beating heart, hydrogel tapes with a diameter of 9 mm were prepared prior to implantation. Then rabbits were anesthetized with isoflurane. The chest hair was removed before the operation, and the regional nerve blocker lidocaine/bupivacaine was injected at the surgical site. A thoracotomy was conducted in the third or fourth left intercostal space to expose the heart. The Electro-Ox hydrogel tape was first adhered to the surface of the heart. Then a metal disk with a diameter of 9 mm and thickness of 1 mm was adhered to the hydrogel tape. The rabbit was ventilated with 100% oxygen for 3 h and then sacrificed by $CO_2$ inhalation.

For hemostasis of a bleeding vessel, pigs (4-month-old male Bama mini pig, n = 5) was anesthetized with Zoletil (Virbac S.A., France). The pigs were not

anticoagulated and the study was not a survival surgery. Then the hair on the pigs' neck was removed, and an incision at the length of ~3 cm was made to expose the jugular vessel. An injury at the length of 2–3 mm was created on a vessel, and blood spouted out at the rate of 6–8 mL min$^{-1}$. Hydrogel tape was pressed to the vessel for 10 s, and the bleeding was immediately stopped. The blood was collected using the medical cotton balls and weighed. For the control group, manual pressure application was used to stop the bleeding. Medical cotton balls were used to press the injury site and the bleeding was stopped for several minutes with rebleeding to certain extent. The weight change of the cotton balls was recorded as the amount of bleeding.

All animal studies were carried out in compliance with the regulations and guidelines of the Ethics Committee of Drum Tower Hospital affiliated to the Medical School of Nanjing University and conducted according to the Institutional Animal Care and Use Committee guidelines.

Statistical significance was determined using Student's $t$ test accordingly with IBM SPSS statistics 26. Statistical significance was set to a $P$ value <0.05.

**Reporting summary**. Further information on research design is available in the Nature Research Reporting Summary linked to this article.

## Data availability
All data are available in the main text, Supplementary Information, and Supplementary Movies. Source data are provided with this paper.

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

## Acknowledgements

This work is supported mainly by the National Key R&D Program of China (Grant No. 2020YFA0908100) to Y.C., the National Natural Science Foundation of China (Nos. 11934008, 11804148, 81730067, 81991514, 21902075, and 12002149) to W.W., B.X., Q.J. and Y.C.; the Natural Science Foundation of Jiangsu Province (No. BK20180320 and BK20200121) to B.X. and L.L.; and the Fundamental Research Funds for the Central Universities (No. 020414380187) to B.X.

## Author contributions

Y.C. and W.W. conceived the idea and designed the study. B.X. and J.G. performed the experiments and analyzed the results. W.Y. prepared some of the hydrogel samples. J.G., L.L., S.Y. and Q.J. performed the in vivo experiments. Y.C., W.W. and B.X. wrote and refined the paper. Y.C., W.W. and M.Q. supervised the project. All the authors discussed the results.

## Competing interests

The authors declare no competing interests.
