## [Peer Review File · Nature Communications]

REVIEWER COMMENTS

Reviewer #1 (Remarks to the Author):

The authors submitted an overall well written manuscript that describes an adhesive tape technology aimed for medical applications including hollow viscous sealing and achieving hemostasis of lacerated blood vessels. Furthermore, the adhesive also provides solid performance on inorganic surfaces. The authors provide in vitro, ex vivo and small and large animal in vivo data. In their manuscript the authors nicely present the underlying mechanism of their adhesive. They provide detailed explanation of the phased adhesion that initially consists of mostly non-covalent interactions of dopa form with substrates from the surrounding. After several hours these bonds will be augmented by covalent bonds of the dopaquinone (oxidized dopa) with primary amine and thiol groups of the surrounding. An interesting aspect of the technology is the 'fail safe' attachment-detachment-reattachment capabilities. This is a very attractive feature and would certainly be welcome by clinicians.

Comments:

They authors illustrate how their technology provides atraumatic adhesive detachment within minutes after application. What about after the covalent bonding has formed? What is the recommended detachment mode, and would it cause tissue injury?

The authors show a compelling sealing performance of a stomach perforation (ex vivo). The authors comment on the pH sensitivity of the adhesion mechanism in nature. Does the ambient pH (e.g. pH 1) affect the adhesion performance? What about presence of enzymes e.g. those from pancreas or bile? Given the target applications of gut, this would be important to know.

The authors further show what appears to be air-tight lung sealing in an ex vivo model but do not show whether this seal lasts for hours of cyclical insufflation (e.g., 20 breaths a minute for 10h with physiological driving pressures and PEEP)

What were the observed burst pressures for stomach and lung seal?

What degree of tissue injury was observed when the tape was torn off after 72h in the rabbit model and what macroscopic/histologic damage – if any – was observed.

The cardiac video runs at 2x which seems unnecessary. How strong was the adhesion after 72h and what macroscopic/histologic damage – if any – was observed.

While the authors provide

The vascular injury model is insufficiently characterized and again does not have a control group (e.g., direct gauze pressure application to achieve hemostasis or use of commercial hemostat etc):

- How many pigs were used?
- How large was the vascular injury?
- What was the bleeding rate (ml/min)?
- Was the pig anticoagulated?
- Could the bleeding be stopped by manual pressure application (going back to control groups)
- Was this a survival surgery?
- Was there rebleeding?
- What was the histologic response of the carotid to the tape application?
- The camera angle is unfortunate and one cannot see the actual application.

The authors claim acceptable biocompatibility but do not appear to provide sham surgery and other control (e.g. commercial tissue adhesive) data as comparator. Moreover, the sole

presentation of TNF alpha data appears insufficient to characterize the immune response comprehensively. TNF will likely be most expressed in the first days after implantation.

Given the time points of 14 and 28 days, one would rather expect collagen deposition (capsule formation?) and macrophage presence to orchestrate bioadhesive breakdown. Again, control groups would be crucial.

More importantly the authors only provide in vivo data of subcutaneous implantation, but no data on the histologic response of the proposed target organs (stomach, lung, artery) to the chronic tape administration.

Reviewer #2 (Remarks to the Author):

Hydrogel tapes for fault-tolerant strong wet adhesion

Here, the authors have prepared a biocompatible and biodegradable mussel-inspired bioadhesive whose adhesion can be controlled via the passage of an electric current. It has been reported that this electro-oxidation of catecholic groups (one of the key adhesion moieties) into quinone becomes progressively stronger with time, while still allowing for repositioning (of the adhesive tape) during the initial moments of exclusive physical interactions.

Major comments:

- If electro-responsive oxidation is the key selling point of this mussel-inspired adhesive, then it is suggested that the authors specify that the adhesive is pre-treated with electric current to induce the formation of quinone groups. Additionally, it would be beneficial to include prior work in which the adhesion of the catechol-containing adhesives is deactivated in situ by the passage of electric current. <https://pubs.acs.org/doi/pdf/10.1021/jacs.9b11266>
- Lines 90-91: Besides the tendency to undergo less undesired side reactions, can the authors also comment on the timescales for chemical oxidation (generally limited by diffusion) and electro-responsive oxidation?
- The authors heavily rely on their HPLC results presented in the supporting information to conclude that electro-oxidation is beneficial for tissue adhesion as compared to chemically-induced oxidation. While the HPLC results look promising, is there any detailed explanation/hypothesis as to 'why' quinone is formed during electro-oxidation vs. the formation of dimer under chemical oxidation?
- Line 206: There are several journal articles that discuss hydrogels with similar chemical composition to the ones presented in this manuscript. How does the adhesion strength of 1268 J/m² compare with similar materials that have been used in the past? The authors must cite the previous papers for the readers' complete understanding.
- Figure 3: The authors have demonstrated adhesion to organic and inorganic substrates. Have they benchmarked their adhesion values with the commercially available products (Tisseel; fibrin-based glue for organic substrates) or cyanoacrylate glue for inorganic surfaces? Also, how does the glue perform against traditionally used sutures (lap shear test)?
- Figure S9: Shear is misspelled on the Y-axis in the big graphs (a) and (b). Also, what is the relationship between oxidation time and adhesion strength? It looks like adhesion decreases drastically beyond 6h of electric treatment.
- In Figure 9, the authors mention using fresh organs, but it appears like the organs have been patted down dry (no blood, bodily fluids; something similar to Figure S11) before applying the adhesive hydrogel. If that is the case, then specific protocols should be mentioned in this case, and appropriate terminology should be used.

Minor comments:

- Line 145 typo: 'and'
- Figure 3: 'Shear' is misspelled again on the Y-axes of (c), (g), (h), and (i).

Reviewer #3 (Remarks to the Author):

This study developed an BSA-Alginate-dopa-PAA hydrogel tape via a controllable electrical oxidation. Owing to the electrical oxidized catecholquinones on the hydrogel, the hydrogel can repeatedly stick to tissue surface initially, and form strong adhesion slowly after hours. This hydrogel tape exhibited strong wet adhesion, including the shear strength, tensile strength and interfacial toughness. The author also demonstrated different potential applications of the BSA-Alginate-dopa-PAA hydrogel. The molecular design of the hydrogel is interesting. However, the advantage of the hydrogel is not impressive enough, which is needed to be promoted.

Specific comments:

1) The dopa can be oxidized in air. During the synthesis process of alginate-dopa, how to prevent the catechols from oxidation.

2) Excitingly, as a hydrogel, the shear and tensile strength of the Electro-Ox hydrogel reached to MPa, which are higher than that of catechol-based hydrogel and catechol-based adhesives or glues, especially under wet condition. It is interesting, please give more discussion on it.

3) Breaking the interfacial hydration layer is critical for wet adhesion. Without the interfacial hydration layer, the adhesive groups can bond with the substrate. In Xuanhe Zhao's work, the PAA-based path is dried, thus their path can adsorb the interfacial water and bond with the substrate. However, in this work, the Electro-Ox hydrogel contained water, which is hard to adsorb interfacial water or break the hydration layer. Thus, the mechanism of wet adhesion in this work should be clearly demonstrated.

4) The study demonstrated that the BSA- alginate-dopa-PAA hydrogel tape is biodegradable. However, most of the components in the hydrogel is PAA, which is not degradable.

5)The advantage design of the study is that with the controllable oxidization, the hydrogel can repeat adhere at the initial time, and form strong adhesion slowly. Compare with the rapid surface covalent bonding, this design can avoid the incorrect adhesion. However, clinically, most applicants require initial strong adhesion. The dynamic environment in body, such as the heartbeat, does not allow slow adhesion on the heart's surface for hours. Moreover, the potential applications in this study are also used the initial adhesion. Thus, the importance of the strong adhesion in the second stage needs to be reconsidered.

6) The wet adhesion of hydrogels is most used for biomedical applications. How much adhesion can meet biomedical applications? In the study, the adhesion of the Chem-Ox and non-OX hydrogel are higher than that of many reported hydrogels. What are their disadvantages in biomedical applications? For examples, can these hydrogels be used in the same applications in this study?

Point-by-point response to the reviewers' comments

Reviewer #1 (Remarks to the Author):

The authors submitted an overall well written manuscript that describes an adhesive tape technology aimed for medical applications including hollow viscous sealing and achieving hemostasis of lacerated blood vessels. Furthermore, the adhesive also provides solid performance on inorganic surfaces. The authors provide in vitro, ex vivo and small and large animal in vivo data. In their manuscript the authors nicely present the underlying mechanism of their adhesive. They provide detailed explanation of the phased adhesion that initially consists of mostly non-covalent interactions of dopa form with substrates from the surrounding. After several hours these bonds will be augmented by covalent bonds of the dopaquinone (oxidized dopa) with primary amine and thiol groups of the surrounding. An interesting aspect of the technology is the 'fail safe' attachment-detachment-reattachment capabilities. This is a very attractive feature and would certainly be welcome by clinicians.

Response:

We thank the reviewer for his/her positive comments.

Comments:

1. They authors illustrate how their technology provides atraumatic adhesive detachment within minutes after application. What about after the covalent bonding has formed? What is the recommended detachment mode, and would it cause tissue injury?

Response:

We thank the reviewer for the comments. After the covalent bonding formed, detachment of the hydrogel tapes can be achieved by adding dithiothreitol (DTT) solutions. The bonds formed between dopaquinone and amine or thiol on tissue surfaces are dynamic covalent bonds. They can be disrupted by the competitive binding of DTT. As shown in Fig. S14, the shear strength of the hydrogel-tape adhered porcine skins decreased by more than 97% (from 1.45 MPa to 0.032 MPa) after being soaked in a PBS solution containing 1 mM DTT for 1 h. This makes it easy to remove the adhered Electro-Ox hydrogel tape. This detaching approach causes ignorable tissue injury. As shown in the SEM images of the porcine skin surfaces after removing the hydrogel-tape, no obvious injury can be observed (Fig. S15). We have added the new comments and data in the revised manuscript. (See the second paragraph on Page 14 and the first paragraph on Page 29 in the revised manuscript; See Fig. S14-15 on Page 13 in the revised Supplementary Information)

Revisions:

...It is worth noting that the detachment of the Electro-Ox hydrogel tapes adhered on the organic tissues can be achieved by adding dithiothreitol (DTT) solutions to the interface. The bonds formed between dopaquinone and amine or thiol on tissue surfaces are dynamic covalent bonds. They can be disrupted by the competitive binding of DTT. As shown in Fig. S14, the shear strength of the hydrogel-tape adhered porcine skins decreased by more than 97% (from 1.45 MPa to 0.032 MPa) after being soaked in a

PBS solution containing 1 mM DTT for 1 h. This makes it easy to remove the adhered Electro-Ox hydrogel tape and causes ignorable tissue injury. As shown in the SEM images of the porcine skin surfaces after removing the hydrogel-tape, no obvious injury can be observed (Fig. S15). ...

Experimental details:

...For the lap shear experiments in the presence DTT solutions, the porcine skin adhered using the hydrogel tapes were soaked in a PBS solution containing 1 mM DTT for 1 h before the shear strength was determined. ...

Fig. S14 Shear strength of the porcine skins adhered using the Electro-Ox hydrogel tape after being treated with DTT solutions (1 mM). **a**, Typical force-displacement curve of the lap shear test. **b**, Comparison of the shear strength of untreated and DTT-treated samples after long-term adhesion was established. Values represent the mean and standard deviation ($n = 5$).

Fig. S15 SEM images of porcine skin surface. **a**, Before the adhesion of the Electro-Ox hydrogel tape. **b**, After removing of the Electro-Ox hydrogel tape. To directly compare the skin surface with and without the hydrogel tape, the hydrogel tape was only partially removed.

2.The authors show a compelling sealing performance of a stomach perforation (ex vivo). The authors comment on the pH sensitivity of the adhesion mechanism in nature. Does the ambient pH (e.g. pH 1) affect the adhesion performance? What about presence of enzymes e.g. those from pancreas or bile? Given the target applications of gut, this would be important to know.

Response:

We thank the reviewer for the comments. The ambient pH would affect the adhesion performance of Electro-Ox hydrogel tapes since the low pH may affect the reaction between quinone and amino/thiol (Fig. S12). In the adhesion test for porcine skins covered a solution at pH 1.0, the long-term adhesion strength (shear strength) was considerably reduced but remained at ~600 kPa. It was still higher than some commercial bio-adhesives at neutral pH.¹

Following the reviewer's suggestion, we also investigated the effects of trypsin from pancreas and raw porcine bile extract on the adhesion of performance of Electro-Ox hydrogel tapes. Trypsin solutions and the bile extract mixture were directly applied to the interface between the hydrogel-tape glued porcine skins and the shear strength was determined after forming long-term adhesion. As shown in Fig. S12, the adhesion strength of porcine skin covered with trypsin solution decreased markedly to 152 kPa. The reduced adhesion strength can be explained by the enzymatic digestion of BSA by trypsin. Similarly, the adhesion strength of porcine skin covered with the bile remained at ~395 kPa. Although bile does not contain enzymes that can digest the hydrogels, the protein components in bile can still partially block the formation of covalent linkage between the hydrogel-tape and the tissue surfaces. Now we have added the new comments and data in the revised manuscript and Supplementary Information. (See line 276-281 on Page 13, the first paragraph on Page 14 and the first paragraph on Page 29 in the revised manuscript; See Fig. S12 on Page 12 in the revised Supplementary Information)

Revisions:

...Besides, the adhesion strength of Electro-Ox hydrogels tapes on organic tissues might also be affected by the ambient pH and enzymes (Fig. S12). The shear strength of the hydrogel-tape adhered porcine skins reduced to ~600 kPa at pH 1.0. The adhesion strength of porcine skin covered with trypsin solution decreased markedly to 152 kPa. The reduced adhesion strength can be explained by the enzymatic digestion of BSA by trypsin. Similarly, the adhesion strength of porcine skin covered with the raw porcine bile extract remained at ~395 kPa. Although bile does not contain enzymes that can digest the hydrogels, the protein components in bile can still partially block the formation of covalent linkage between the hydrogel-tape and the tissue surfaces. Note that even the adhesion strength was much reduced at low pH or in the presence of enzymes, it was still comparable with or even higher than the adhesion strength of some commercial bio-adhesives such as polyethylene-glycol and fibrinogen-based tissue adhesives.²⁹...

Experimental details:

...For the lap shear experiments in the presence of trypsin, bile or at low pH, the solution of trypsin (2.5 mg mL⁻¹ in PBS), the raw porcine bile extract or a solution at pH 1.0 were applied to the porcine skin surface before the adhesion...

Fig. S12 Adhesion of porcine skins using the Electro-Ox hydrogel tape in the presence of bile/trypsin or a low pH solution (pH=1.0). **a**, Typical force-displacement curve of the lap shear test. **b**, Shear strength measured at low pH or with enzymes. Values represent the mean and standard deviation ($n = 5$).

3. The authors further show what appears to be air-tight lung sealing in an ex vivo model but do not show whether this seal lasts for hours of cyclical insufflation (e.g., 20 breaths a minute for 10h with physiological driving pressures and PEEP)

Response:

We thank the reviewer for the comments. The air-tight lung sealing in Fig. 4c and d was used to suggest the instant sealing of the lung. A constant pressure was applied to the sealed lung instead of using the cyclical insufflation. Following the reviewer's suggestion, we now show the sealing of a lung under cyclical insufflation. As shown in Fig. S19, the test was driven by a ventilator (MEDUMAT Standard², WEINMANN, Germany) at the mode of Intermittent Positive Pressure Ventilation (IPPV). The hydrogel tape was colored with a dark food dye for visualization. The tidal volume (V_t), frequency, maximum pressure (p_{Max}), and positive end expiratory pressure (PEEP) were first set as 275 mL, 13 breaths per minute, 15 mBar, and 5 mBar, respectively. The peak pressure (p_{Peak}) was 14 mBar. Then an injury (0.5×1.5 cm) was created to the lung and the p_{Peak} decreased to 12 mBar. After the sealing of the lung using an Electro-Ox hydrogel tape (diameter of 4.0 cm), the p_{Peak} immediately recovered back to 14 mBar under the cyclical insufflation, indicating the fast and stable sealing of lung. Then the V_t and p_{Max} were changed to 600 mL and 40 mBar to increase the pressure applied on the Electro-Ox hydrogel tape. As indicated by Fig. S19, the p_{Peak} remained at 21-22 mBar even after 8 h, indicating the long-term stability of the sealing of lung using the Electro-Ox hydrogel tape. The lung and hydrogel were sprayed with PBS solution (10 mM, pH=7.4) to keep wet during the test. Now we have added the new results and experimental details in the revised manuscript. (See the second paragraph on Page 18 in the revised manuscript; See the fourth paragraph on Page 4 and Fig. S19 on Page 16 in the revised Supplementary Information)

Revisions:

...Moreover, the peak pressure of sealed lung can remain stable under cyclical insufflations lasting for 8 h (Frequency: 13 breathe per minute; tidal volume (V_t): 600 mL; maximum pressure (p_{Max}): 40 mBar; positive end expiratory pressure (PEEP): 5 mBar), suggesting the robustness of the lung sealing (Fig. S19). ...

Experimental details:

...For the sealing of a lung under the cyclical insufflation, the contraction and relaxation of the lung was driven by a ventilator (MEDUMAT Standard², WEINMANN, Germany) at the mode of Intermittent Positive Pressure Ventilation (IPPV). The hydrogel tape is colored with a dark food dye for visualization. The tidal volume (V_t), frequency, maximum pressure (p_{Max}), and positive end expiratory pressure (PEEP) were first set as 275 mL, 13 breaths per min, 15 mBar, and 5 mBar, respectively. Then an injury (0.5×1.5 cm) was created to the lung. After sealing the lung using an Electro-Ox hydrogel tape (diameter of 4.0 cm), the V_t and p_{Max} were changed to 600 mL and 40 mBar to increase the pressure applied to the lung. The cyclical insufflation lasted for 8 h. ...

Fig. S19 Air-tight lung sealing under the cyclical insufflation using the Electro-Ox hydrogel tape. The value of p_{peak} recovered from 12 mBar of the injured lung to 14 mBar of the sealed lung, which was the same as that of the uninjured lung. After increasing the p_{Max} and V_t to 40 mBar and 600 mL, the p_{peak} increased to 21 mBar and remained stable at the range of 21-22 mBar in the following 8 h of cyclical insufflation.

The lung and hydrogel were sprayed with PBS solution (10 mM, pH=7.4) to keep wet during the test.

4. What were the observed burst pressures for stomach and lung seal?

Response:

We thank the reviewer for the comments. The burst pressures for stomach and lung seal were measured using a piezometer (Fig. S20). The observed burst pressures for short-term sealing of stomach and lung were ~20.6 and ~17.9 kPa, respectively. These values then increased to ~69.0 and ~58.0 kPa for long-term sealing. The high burst pressures indicated the outstanding sealing strength using the Electro-Ox hydrogel tapes. Now we have added the new results and experimental details in the revised manuscript. (See line 374-375 on Page 18 in the revised manuscript; See the second paragraph on Page 5 and Fig. S20 on Page 17 in the revised Supplementary Information)

Revisions:

... The observed burst pressures for short-term sealing of stomach and lung were ~20.6 and ~17.9 kPa, respectively (Fig. S20). These values then increased to ~69.0 and ~58.0 kPa for long-term sealing. ...

Experimental details:

... For the evaluation of burst pressures, the stomach or lung tissue with a thickness between 3-5 mm were cut off from the porcine organs. The tissues were fixed on a pumping chamber and a penetrating defect (diameter of 4 mm) was created at the center of the tissue. A hydrogel tape (10 mm×10 mm) was applied to the defect for 15 s (short-term sealing) or 24 h (long-term sealing). The pressure was increased by pumping PBS into the chamber at the speed of 2 mL min⁻¹ with a syringe pump (WZS-50F6, Smiths Medical, China). The burst pressure during the test was recorded using a digital piezometer (SW-512C, SNDWAY, China). ...

Fig. S20 Burst pressure of the porcine stomach and lung sealed by the Electro-Ox

hydrogel tapes. a, Schematic illustration of the burst pressure measurement using the punctured porcine tissues. **b**, Typical images of the burst pressure measurements using the sealed porcine stomach as an example. **c**, Burst pressures of short-term and long-term sealing for the porcine stomach and lung. Values represent the mean and standard deviation ($n = 4$).

5. What degree of tissue injury was observed when the tape was torn off after 72h in the rabbit model and what macroscopic/histologic damage – if any – was observed. The cardiac video runs at $2\times$ which seems unnecessary. How strong was the adhesion after 72h and what macroscopic/histologic damage – if any – was observed.

Response:

We thank the reviewer for the comments. Following the reviewer's suggestion, we studied the tissue injury when the tape was torn off after 72 h in the rabbit model. The Electro-Ox hydrogel tapes were removed from the gastrocnemius muscle of rabbits and the injury was estimated by histological assessments. As shown in Fig. S22a, the tissues did not show obvious injury, indicating the negligible damage caused by tearing off the Electro-Ox hydrogel tapes.

The cardiac video now runs at $1\times$ speed as the reviewer suggested (See Movie S5). The adhesion strength after 72 h remained at ~ 80 kPa as indicated by the shear strength test (Fig. S21). The injury of heart was also evaluated by histological assessments as shown in Fig. S22b. Similarly, no obvious damage could be observed for the heart tissue. We have also included these results in the revised manuscript. (See the first paragraph on Page 20 in the revised manuscript; See Fig. S21 on Page 17 and Fig. S22 on Page 18 in the revised Supplementary Information)

Revisions:

...After the adhesion for 72 h, the shear strength was ~ 80 kPa as indicated by the lap shear test (Fig. S21). However, even the hydrogel tapes were removed from the gastrocnemius or heart at this stage, no obvious injuries were observed, suggesting that the covalent bonding of hydrogel-tapes to the tissue surfaces did not cause deep tissue injury (Fig. S22a and b). ...

Fig. S21 Adhesion strength of the hydrogel-tape applied to the rabbit heart in vivo. **a**, Typical force-displacement curves of the lap shear test. **b**, Summary of the shear strength. The “initial” group is the shear strength after long-term adhesion was established and the “72 h” group is the shear strength after in vivo adhesion for 72 h.

Values represent the mean and standard deviation (n = 5).

Fig. S22 Representative H&E staining of the tissues in the in vivo adhesion and haemostasis using the Electro-Ox hydrogel tapes. a, Representative H&E staining of rabbit gastrocnemius after in vivo adhesion with an Electro-Ox hydrogel tape for 72 h and then the hydrogel was torn off. The sham surgery was set as the control group. **b,** Representative H&E staining of rabbit heart after in vivo adhesion of Electro-Ox hydrogel tape for 72 h and then the hydrogel was torn off. The normal heart was set as the control group. **c,** Representative H&E staining of porcine vessel after in vivo adhesion of Electro-Ox hydrogel tape for 72 h and then the hydrogel was torn off. The sham surgery was set as the control group.

6. While the authors provide

The vascular injury model is insufficiently characterized and again does not have a control group (e.g., direct gauze pressure application to achieve hemostasis or use of commercial hemostat etc):

- How many pigs were used?
- How large was the vascular injury?
- What was the bleeding rate (ml/min)?
- Was the pig anticoagulated?
- Could the bleeding be stopped by manual pressure application (going back to control groups)
- Was this a survival surgery?
- Was there rebleeding?
- What was the histologic response of the carotid to the tape application?
- The camera angle is unfortunate and one cannot see the actual application.

Response:

We thank the reviewer for the comments. Following the reviewer's suggestions, we reperformed the haemostasis experiments including experimental and control groups using the vascular injury model. Five pigs were used in the experiments. The experiment about the vascular injury was not a survival surgery. Typically, an injury at the length of 2-3 mm was created on a vessel and the bleeding rate was about 6-8 mL min⁻¹. The pigs were not anticoagulated. The pigs whose bleeding was stopped by manual pressure using medical cotton balls were set as the control. It took ~9 min to stop bleeding by manual pressure using medical cotton balls (Fig. S23b). In contrast, the bleeding could be stopped immediately after applying the Electro-Ox hydrogel tape and no obvious rebleeding was observed in 15 min after the stop of bleeding (Fig. S23a and movie S6). Moreover, the amount of bleeding using the hydrogel tape was less than 10% of that by manual pressure using medical cotton balls (Fig. S23c). The video of the haemostasis for the bleeding artery was retaken with a suitable camera angle for observation (Movie S6).

The histologic response of the artery to the hydrogel tape was also investigated (Fig. S22c). In this case the hydrogel was applied to an uninjured carotid vein to avoid the effects of blood coagulation. No obvious tissue damage or pathological change was observed after applying the hydrogel tape for 72 h, indicating the hydrogel tapes are suitable for haemostasis.. We have added the new results in the revised manuscript. (See line 408-412 on Page 20, line 413-415 and Fig. 5f on Page 21, and the first paragraph on Page 33 in the revised manuscript; See Fig. S22 on Page 18 and Fig. S23 on Page 19 in the revised Supplementary Information; See Movie S6 in the Supplementary videos)

Revisions:

...After gently compressing the Electro-Ox hydrogel tape on a bleeding carotid artery, the bleeding was stopped without any leakage and no rebleeding was observed in the following 15 min (Fig. 5f, S23a and Movie S6). Comparing to the haemostasis using manual pressure, the amount of bleeding using the hydrogel tape decreased by more than 90% (Fig. S23b and c). The histologic response of the artery to the hydrogel tape was also investigated (Fig. S22c). No obvious tissue damage or pathological change was observed after applying the hydrogel tape for 72 h, indicating the hydrogel tapes are suitable for haemostasis. ...

Experimental details:

...For haemostasis of a bleeding vessel, pigs (Bama mini pig, n=5) was anaesthetized with Zoletil (Virbac S.A., France). The pigs were not anticoagulated and the study was not a survival surgery. Then, the hair on the pigs' neck was removed, and an incision at the length of ~3 cm was made to expose the jugular vessel. An injury at the length of 2-3 mm was created on a vessel, and blood spouted out at the rate of 6-8 mL min⁻¹. Hydrogel tape was pressed to the vessel for 10 s, and the bleeding immediately stopped. The blood was collected using the medical cotton balls and weighed. For the control group, the manual pressure application was used to stop the bleeding. The medical

cotton balls were used to press the injury site and the bleeding stopped in several minutes with rebleeding to certain extents. The weight change of the cotton balls was recorded as the amount of bleeding....

Fig. S23 Haemostasis of the vascular injury of pigs. a, Haemostasis of the vascular injury of pigs using Electro-Ox hydrogel tapes in vivo. **b,** Haemostasis of the vascular injury of pigs by manual pressure using medical cotton balls in vivo. **c,** Comparison of the amounts of bleeding using different methods. Values represent the mean and standard deviation (n = 5).

Fig. S22 Representative H&E staining of the tissues in the in vivo adhesion and haemostasis using the Electro-Ox hydrogel tapes. **a**, Representative H&E staining of rabbit gastrocnemius after in vivo adhesion with an Electro-Ox hydrogel tape for 72 h and then the hydrogel was torn off. The sham surgery was set as the control group. **b**, Representative H&E staining of rabbit heart after in vivo adhesion of Electro-Ox hydrogel tape for 72 h and then the hydrogel was torn off. The normal heart was set as the control group. **c**, Representative H&E staining of porcine vessel after in vivo adhesion of Electro-Ox hydrogel tape for 72 h and then the hydrogel was torn off. The sham surgery was set as the control group.

7. The authors claim acceptable biocompatibility but do not appear to provide sham surgery and other control (e.g. commercial tissue adhesive) data as comparator. Moreover, the sole presentation of TNF alpha data appears insufficient to characterize the immune response comprehensively. TNF will likely be most expressed in the first days after implantation.

Response:

We thank the reviewer for the comments. Now we have reperformed the experiments of subcutaneous implantation in the backs of Sprague Dawley rats including the experimental groups, the control groups of the sham surgery and commercial tissue adhesives (fibrinogen-based and cyanoacrylate-based bio-adhesives) and the groups without surgery. As shown by the histologic assignment, moderate acute inflammatory responses were found in all surgery groups on the first day and the inflammatory reactions almost disappeared after 14 d (Fig. 6d-g and S25). The immunofluorescence staining of TNF- α and IL-1 β staining for the samples after subcutaneous implantation in the backs of Sprague Dawley rats for one day were also presented (Fig. S26). A low expression level of TNF- α and IL-1 β in tissues around the hydrogel tapes was observed

similar to the effect of commercially available bio-adhesives and sham surgery. These results indicated the moderate inflammatory responses of the Electro-Ox hydrogels comparable to those of the commercially available bio-adhesives and sham surgery. Furthermore, the examinations of blood biochemistry (ALT, AST, UREA, CREA, Ca²⁺ and K⁺), cell (WBC, RBC, HGB, and LY), and coagulation (APTT and FIB) were performed after implantation for different days (Fig. 6h-s). All the blood examination indicators did not change obviously after the Electro-Ox hydrogel was implanted for different days. The similar values of the indicators as those of the commercially available bio-adhesives and sham operation demonstrated that the implantation of the hydrogel tape did not affect the function of heart, liver, and kidney, as well as the coagulation of blood. The new data and comments are now included in the revised manuscript. (See the second paragraph on Page 22, the whole Page 23, Fig. 6 on Page 25 and the third paragraph on Page 30 in the revised manuscript; See Fig. S25-26 on Page 20 in the revised Supplementary Information)

Revisions:

...We finally evaluated the *in vivo* biocompatibility and biodegradability of the tape based on dorsal subcutaneous implantation in a rat model (Fig. 6d-s and S25). The groups of the sham surgery and commercial tissue adhesives (fibrinogen-based and cyanoacrylate-based bio-adhesives) as well as the groups without surgery were also investigated for comparison. As indicated by the histological assessment (hematoxylin and eosin (H&E) staining) results, the hydrogel tapes did not cause any major damage to the surrounding dermal and muscular layers (Fig. 6d). A mild chronic inflammatory response could be found in all the groups on the first day, as indicated by the presence of lymphocytes cells and moderate eosinophilic response (Fig. 6d-g). No necrosis of the skeletal muscle and skin was observed. The inflammatory response was considerably weakened with increasing time and almost disappeared after 14 days. The immunofluorescence staining of TNF- α and IL-1 β staining for the samples in 1 day after subcutaneous implantation in the backs of Sprague Dawley rats were also presented (Fig. S25 and S26). A low expression level of TNF- α and IL-1 β around the hydrogel tapes were observed similar to those of the commercially available bio-adhesives and sham surgery. Moreover, the Electro-Ox hydrogel tapes exhibited fast degradation in 14 days, similar to that of the fibrinogen-based bio-adhesives (Fig. 6d and e). Formation of collagenous capsules and granulation tissue around the implantation sites was observed after 14 days, indicating the breakdown of the hydrogel tapes and the healing of the damaged tissues. In contrast, the cyanoacrylate-based bio-adhesive did not degrade in 14 days after implantation (Fig. 6f).

Furthermore, we conducted systemic toxicity tests to evaluate the generalized biological effects of the hydrogel tape. The blood biochemistry (ALT, AST, UREA, CREA, Ca²⁺ and K⁺), cell (WBC, RBC, HGB, and LY), and coagulation (APTT and FIB) indicators did not show obvious change after implantation (Fig. 6h-s). The similar values of the indicators as those of the commercially available bio-adhesives and sham operation demonstrated that the implantation of the hydrogel tape did not affect the

function of heart, liver, and kidney, as well as the coagulation of blood. ...

Experimental details:

...For in vivo biocompatibility, hydrogel tapes and fibrinogen-based bio-adhesives with the width of 8 mm and length of 8 mm were prepared and stored in a mixture of alcohol (75%) and ddH₂O (25%) prior to implantation. Then, Sprague Dawley rats aged 8 weeks (250±10 g) were randomly divided into four groups (n=15). (group A: the Electro-Ox hydrogel was administered; group B: the fibrinogen-based bio-adhesive was administered; group C: the cyanoacrylate-based bio-adhesive was administered; and group D: the sham surgery was performed). The rats were anaesthetized with isoflurane. The back hair of the rats was removed, and each sample was inserted between the skin and muscle through a 1.5-cm skin incision at the center of the rat back. Then, the incision was closed using interrupted sutures (3-0 Vicryl Plus, JNJ). After 1, 7 or 14 days, 5 of the rats were sacrificed, and target subcutaneous regions as well as major organs were obtained and fixed in 10% formalin for 24 h before histological analysis (H&E staining) and immunofluorescent staining (TNF- α and IL-1 β). Blood samples were collected at the same time points for the systemic toxicity tests. ...

Fig. 6 Biocompatibility and biodegradability of Electro-Ox hydrogel tapes in vitro and in vivo. **a, b**, Fluorescence microscopy images (**a**) and cell viability (**b**) of MC 3T3 and MEF cells cultured on Electro-Ox hydrogels. The living and dead cells were stained with a live/dead assay (Calcein-AM/PI Double Staining Kit) after 24 h of culture. NS: $p > 0.05$. Scale bar = 100 μm . **c**, In vitro biodegradation of Electro-Ox hydrogel tape in SBF with or without protease. Values in **b-c** represent the mean and standard deviation ($n = 3-5$). **d**, Representative histological images stained with haematoxylin and eosin (H&E) for assessment of the biocompatibility and

biodegradation of the Electro-Ox hydrogel tape in vivo at Day 1, 7, and 14 after subcutaneous implantation. GT and FC indicate granulation tissue and fibrous capsule, respectively. **e-g**, Representative histological images of the control groups. **(e)** the fibrin-based glue, **(f)** the cyanoacrylate glue, and **(g)** the sham surgery. SM and GT indicate skeletal muscle and granulation tissue, respectively. All histological experiments were repeated at least three times with similar results. Scale bar = 100 μm . **h-m**, Blood biochemistry examination of the variation in ALT (alanine transferase), AST (aspartate transferase), UREA (urea), CREA (creatinine), Ca^{2+} , and K^+ . **n-q**, Hematological examination of the variation in WBC (white blood cell count), RBC (red blood cell count), HGB (hemoglobin), and LY (lymphocyte). **r, s**, Coagulation action test of the variation in APTT (activated partial thromboplastin time) and FIB (fibrinogen). The blood of rats without any surgery was set as the control. In **h-s**, the samples were taken at Day 1, 7, and 14 after subcutaneous implantation and reported values represent the mean and standard deviation ($n = 3$).

Fig. S25 H&E and inflammatory factor staining of the subcutaneous tissues from rats without surgery. a, Representative image of the H&E staining. **b, c**, Representative immunostaining images of TNF- α (**b**) and IL-1 β (**c**) of the subcutaneous tissues from rats without surgery. Cell nuclei are indicated by DAPI (Blue). No inflammatory effect can be observed on the tissues without surgery. Scale bar=100 μm .

Fig. S26 Evaluation of the inflammatory responses in the first day after the subcutaneous implantation. a, Representative immunostaining images identified by molecular markers (TNF- α or IL-1 β , colored in red). Cell nuclei are indicated by DAPI (blue). Scale bar=100 μ m. **b-c**, Relative intensity of TNF- α and DAPI (b) or IL-1 β and DAPI (c) in the images for various samples and the one without surgery was set as the control. Values represent the mean and the standard deviation (n = 5).

8. Given the time points of 14 and 28 days, one would rather expect collagen deposition (capsule formation?) and macrophage presence to orchestrate bioadhesive breakdown. Again, control groups would be crucial.

Response:

We thank the reviewer for the comments. According to the new results in Fig. 6d-g, the Electro-Ox hydrogel tapes exhibited fast degradation in 14 days, similar to that of the fibrinogen-based bio-adhesives. Formation of collagenous capsules and granulation tissue around the implantation sites was observed after 14 days, indicating the breakdown of the hydrogel tapes and the healing of the damaged tissues. In contrast, the cyanoacrylate-based bio-adhesive did not degrade in 14 days after implantation (Fig. 6f). The control groups of sham surgery and commercial adhesives were provided in the new experiments. The data and new comments are now included in the revised manuscript. (See line 458-463 on Page 23 and Fig. 6 on Page 25 in the revised manuscript)

Revisions:

... Moreover, the Electro-Ox hydrogel tapes exhibited fast degradation in 14 days, similar to that of the fibrinogen-based bio-adhesives (Fig. 6d and e). Formation of collagenous capsules and granulation tissue around the implantation sites was observed after 14 days, indicating the breakdown of the hydrogel tapes and the healing of the damaged tissues. In contrast, the cyanoacrylate-based bio-adhesive did not degrade in 14 days after implantation (Fig. 6f)....

Fig. 6 Biocompatibility and biodegradability of Electro-Ox hydrogel tapes in

vitro and in vivo. a, b, Fluorescence microscopy images (**a**) and cell viability (**b**) of MC 3T3 and MEF cells cultured on Electro-Ox hydrogels. The living and dead cells were stained with a live/dead assay (Calcein-AM/PI Double Staining Kit) after 24 h of culture. NS: $p > 0.05$. Scale bar = 100 μm . **c**, In vitro biodegradation of Electro-Ox hydrogel tape in SBF with or without protease. Values in **b-c** represent the mean and standard deviation ($n = 3-5$). **d**, Representative histological images stained with haematoxylin and eosin (H&E) for assessment of the biocompatibility and biodegradation of the Electro-Ox hydrogel tape in vivo at Day 1, 7, and 14 after subcutaneous implantation. GT and FC indicate granulation tissue and fibrous capsule, respectively. **e-g**, Representative histological images of the control groups. (**e**) the fibrin-based glue, (**f**) the cyanoacrylate glue, and (**g**) the sham surgery. SM and GT indicate skeletal muscle and granulation tissue, respectively. All histological experiments were repeated at least three times with similar results. Scale bar = 100 μm . **h-m**, Blood biochemistry examination of the variation in ALT (alanine transferase), AST (aspartate transferase), UREA (urea), CREA (creatinine), Ca^{2+} , and K^+ . **n-q**, Hematological examination of the variation in WBC (white blood cell count), RBC (red blood cell count), HGB (hemoglobin), and LY (lymphocyte). **r, s**, Coagulation action test of the variation in APTT (activated partial thromboplastin time) and FIB (fibrinogen). The blood of rats without any surgery was set as the control. In **h-s**, the samples were taken at Day 1, 7, and 14 after subcutaneous implantation and reported values represent the mean and standard deviation ($n = 3$).

9. More importantly the authors only provide in vivo data of subcutaneous implantation, but no data on the histologic response of the proposed target organs (stomach, lung, artery) to the chronic tape administration.

Response:

We thank the reviewer for the comments. The histologic response of the major organs (heart, liver, spleen, lung, kidney, stomach and artery) to the chronic tape administration in the rat model was studied following the reviewer's suggestion. As shown in Fig. S27 and S28, no obvious histologic response or pathological change to the major organs (heart, liver, spleen, lung, kidney, stomach and artery) was observed in 14 days after implantation. The new data and comments are now included in the revised manuscript. (See the second paragraph on Page 24 in the revised manuscript; See Fig. S27-28 on Page 21-22 in the revised Supplementary Information)

Revisions:

...At last, we further evaluated the histologic response of the major organs (heart, liver, spleen, lung, kidney, stomach and artery) to the chronic administration of the Electro-Ox hydrogel and other bio-adhesives (Fig. S27 and S28). No obvious histologic response or pathological change to the major organs (heart, liver, spleen, lung, kidney, stomach and artery) was observed in 14 days after implantation. ...

Fig. S27 Representative H&E staining of major organs (the heart, liver, spleen, lung, kidney, stomach and artery) in 1 day after subcutaneous implantation of different bio-adhesives in the backs of Sprague Dawley rats. The organs from the rats without surgery were set as the control group.

Fig. S28 Representative H&E staining of major organs (the heart, liver, spleen, lung, kidney, stomach and artery) at the time point of 14 days after subcutaneous implantation of different bio-adhesives in the backs of Sprague Dawley rats. The organs from the rats without surgery were set as the control group.

Reviewer #2 (Remarks to the Author):

Hydrogel tapes for fault-tolerant strong wet adhesion

Here, the authors have prepared a biocompatible and biodegradable mussel-inspired bioadhesive whose adhesion can be controlled via the passage of an electric current. It has been reported that this electro-oxidation of catecholic groups (one of the key adhesion moieties) into quinone becomes progressively stronger with time, while still allowing for repositioning (of the adhesive tape) during the initial moments of exclusive physical interactions.

Response:

We thank the reviewer for these comments.

Major comments:

1. If electro-responsive oxidation is the key selling point of this mussel-inspired adhesive, then it is suggested that the authors specify that the adhesive is pre-treated with electric current to induce the formation of quinone groups. Additionally, it would be beneficial to include prior work in which the adhesion of the catechol-containing

adhesives is deactivated in situ by the passage of electric current. <https://pubs.acs.org/doi/pdf/10.1021/jacs.9b11266>

Response:

We thank the reviewer for the comments. We have specified that the alginate-dopa is pre-treated with electric current to induce the formation of quinone groups. We have also cited the prior work reviewer recommended in the revised manuscript. (See reference No.56 and the first paragraph on Page 5 in the revised manuscript)

Revisions:

...The alginate-dopa was partially electrically oxidized using a galvanic cell prior to hydrogel preparation (Fig. S2). Electro-oxidation ensured that most of the products were dopaquinone, which was in sharp contrast to the chemical oxidation products (Fig. S3a-e). The chemical oxidation of dopa by potassium periodate (KIO₄) mainly led to the formation of catechol dimers that cannot form covalent bonds with amino or thiol groups. ...

2. Lines 90-91: Besides the tendency to undergo less undesired side reactions, can the authors also comment on the timescales for chemical oxidation (generally limited by diffusion) and electro-responsive oxidation?

Response:

We thank the reviewer for the comments. Generally, the timescale of chemical oxidation is significantly shorter than that of electro-responsive oxidation. As shown in Fig. S3b, the chemical oxidation of N-acetyldopamine by KIO₄ and electrical oxidation of N-acetyldopamine under a voltage of 5 V were both performed for one hour. However, the HPLC traces indicated that the conversion of dopa to oxidation products was only ~40% in the electro-oxidation process while the yield of chemical-oxidation was ~80%, suggesting much faster oxidation speed of chemical oxidation. Unlike chemical oxidation, which is generally limited by diffusion, electrochemical oxidation is an electron charge transfer process occurred at the anode surface. The reaction rate can be affected by many factors, including electrolyte resistance of the electrode, the surface adsorption of reactants, and the mass transfer of the reactants from bulk solution to the electrode surface. It is possible to improve the electrochemical oxidation speed in the future by optimizing the surface properties of the electrode. We have added the new comments in the revised manuscript. (See the first paragraph on Page 5 in the revised manuscript; See Fig. S3 on Page 7 in the Supplementary Information)

Revisions:

...However, electrochemical oxidation was slower than the chemical oxidation process. The electrochemical oxidation rate depends on many factors, including electrolyte resistance of the electrode, the surface adsorption of reactants, and the mass transfer of the reactants from bulk solution to the electrode surface. It is possible to improve the electrochemical oxidation speed in the future by optimizing the surface properties of the electrode. Nonetheless, the electro-oxidation method allowed us to selectively and controllably oxidize dopa to dopaquinone without producing other side products (Fig.

S3f-i). ...

3. The authors heavily rely on their HPLC results presented in the supporting information to conclude that electro-oxidation is beneficial for tissue adhesion as compared to chemically-induced oxidation. While the HPLC results look promising, is there any detailed explanation/hypothesis as to 'why' quinone is formed during electro-oxidation vs. the formation of dimer under chemical oxidation?

Response:

We thank the reviewer for the comments. The electro-oxidation of dopa on electrodes has been studied by many researchers²⁻⁴ (Akram Bhuiyan MS, *et al. J Am Chem Soc* **142**, 4631-4638 (2020); Lakshmi D, *et al. Anal Chem* **81**, 3576-3584 (2009); Mao A, *et al. Talanta* **144**, 252-257 (2015)). For the electro-oxidation of dopa adsorbed on the electrode, dopa is first converted into dopaquinone. When dopaquinone on the electrode surface reaches a high concentration, polymerization/crosslinking reaction of dopa and dopaquinone takes place. The crosslinking reaction requires that two dopa/dopaquinone molecules are adsorbed in proximity on the electrode, which is difficult to achieve practically. Therefore, the dominant product is dopaquinone. In contrast, the chemical oxidation takes place in solution, in which dopa and dopaquinone diffuse freely, inevitably leading to the formation of dimer. We have added the discussion in the revised manuscript. (See the end of Page 5 and the first paragraph on Page 6 in the revised manuscript)

Revisions:

... The electro-oxidation of dopa on electrodes has been studied by many researchers⁵⁶⁻⁵⁸. For electro-oxidation of dopa adsorbed on the electrode, dopa is first converted into dopaquinone. When dopaquinone on the electrode surface reaches a high concentration, polymerization/crosslinking reaction of dopa and dopaquinone takes place. The crosslinking reaction requires that two dopa/dopaquinone molecules are adsorbed in proximity on the electrode, which is difficult to achieve practically. Therefore, the dominant product is dopaquinone. In contrast, the chemical oxidation takes place in solution, in which dopa and dopaquinone diffuse freely, inevitably leading to the formation of dimer. ...

4. Line 206: There are several journal articles that discuss hydrogels with similar chemical composition to the ones presented in this manuscript. How does the adhesion strength of 1268 J/m² compare with similar materials that have been used in the past? The authors must cite the previous papers for the readers' complete understanding.

Response:

We thank the reviewer for the comments. The adhesion strength of 1268 J/m² was significantly higher than that of the materials with similar chemical composition (10-280 J/m²).⁵⁻¹¹ Now we have cited the previous papers following the reviewer's suggestion. (See the second paragraph on Page 11, reference 48 and 63-68 in the revised manuscript)

Revisions:

The interfacial toughness of the Electro-Ox hydrogel tape and the porcine skin reached 1268 J m^{-2} , outperforming many reported bio-adhesives.^{48, 63-68}

5. Figure 3: The authors have demonstrated adhesion to organic and inorganic substrates. Have they benchmarked their adhesion values with the commercially available products (Tisseel; fibrin-based glue for organic substrates) or cyanoacrylate glue for inorganic surfaces? Also, how does the glue perform against traditionally used sutures (lap shear test)?

Response:

We thank the reviewer for the comments. Now we have presented the comparison between Electro-Ox hydrogel tape and commercially available products as the reviewer recommended. As shown in Fig. S16, the shear strength for long-term adhesion of fibrin-based bio-glue (SHRAAS, Shanghai, China) on organic substrates ($\sim 12.6 \text{ kPa}$ for vessel, $\sim 19.6 \text{ kPa}$ for stomach, $\sim 28.6 \text{ kPa}$ for liver, $\sim 6.3 \text{ kPa}$ for intestine, and $\sim 8.8 \text{ kPa}$ for heart) was significantly lower than that of Electro-Ox hydrogel tapes ($> 100 \text{ kPa}$). Meanwhile, the shear strength of the long-term adhesion for cyanoacrylate glue (Compant. Beijing, China) on inorganic substrates ($\sim 2.9 \text{ MPa}$ for Fe, $\sim 3.6 \text{ MPa}$ for SiO_2 , $\sim 8.0 \text{ MPa}$ for PMMA and $\sim 3.6 \text{ MPa}$ for glass) were much higher than those of the Electro-Ox hydrogels. We also studied the breaking strength of porcine skin cured using traditional sutures (3-0 Vicryl Plus, JNJ). As shown in Fig. S9, the breaking strength from the lap shear test was $\sim 2.0 \text{ MPa}$, comparable to that of the Electro-Ox hydrogel tapes ($\sim 1.46 \text{ MPa}$). All these results indicated the organic adhesion ability of the Electro-Ox hydrogel tapes is comparable with or even superior to many commercially available adhesives and sutures. We have included the comparison in the revised manuscript. (See the second paragraph on Page 11, the second paragraph on Page 15 and the second paragraph on Page 16 in the revised manuscript; See Fig. S9 on Page 10 and S16 on Page 14 in the revised Supplementary Information)

Revisions:

...As summarized in Fig. 3f, the shear and tensile strength between the Electro-Ox hydrogel tape and the wet porcine skin were $\sim 1.46 \text{ MPa}$ and 1.25 MPa , which are comparable with that of commonly used sutures (Fig. S9, $\sim 2.0 \text{ MPa}$). ...

...The shear strengths of the Electro-Ox hydrogel tapes for different tissues were significantly higher than those of fibrinogen-based bio-adhesives (Fig. S16a and c), which are usually less than 30 kPa ($\sim 12.6 \text{ kPa}$ for vessel, $\sim 19.6 \text{ kPa}$ for stomach, $\sim 28.6 \text{ kPa}$ for liver, $\sim 6.3 \text{ kPa}$ for intestine, and $\sim 8.8 \text{ kPa}$ for heart). ...

...Even though the shear strength of the Electro-Ox hydrogel tapes on these solid surfaces were smaller than those of cyanoacrylate-based glue (Fig. S16b and d; $\sim 2.9 \text{ MPa}$ for Fe, $\sim 3.6 \text{ MPa}$ for SiO_2 , $\sim 8.0 \text{ MPa}$ for PMMA and $\sim 3.6 \text{ MPa}$ for glass), the adhesion strength is strong enough for fixing electronic devices.³⁵...

Fig. S9 Breaking strength of porcine skins jointed by traditional sutures. a, Typical force-displacement curve. **b,** Comparison of the breaking strength for the porcine skins jointed using Electro-Ox hydrogel tapes and sutures. Values represent the mean and standard deviation ($n = 5$).

Fig. S16 Adhesion strengths of commercially available bio-adhesives. a, Typical force-displacement curves of the lap shear test for the adhesion of different porcine organs using the fibrinogen-based bio-adhesive. **b,** Typical force-displacement curves of the lap shear test for the adhesion of different substrates using the cyanoacrylate-based glue. **c,** Summary of the adhesion strength for different porcine organs using the fibrinogen-based bio-adhesive. **d,** Summary of the adhesion strength for different substrates using the cyanoacrylate-based glue. The glued samples were placed at room temperature for at least 24 h to ensure the adhesion strength reached the maximum. Values represent the mean and standard deviation ($n=5$).

6. Fig. S9: Shear is misspelled on the Y-axis in the big graphs (a) and (b). Also, what is the relationship between oxidation time and adhesion strength? It looks like adhesion

decreases drastically beyond 6h of electric treatment.

Response:

We thank the reviewer for the comments. Now we have corrected the misspelling in Fig. S10. The adhesion strength of the Electro-Ox hydrogel tapes first increased with the increase of the oxidation time in 6 h and then decreased gradually afterwards. We inferred that the dopaquinone groups on alginate reached the maximum at the oxidation time of 6 h. Meanwhile, still quite some dopa remained unoxidized and they contributed to the short-term adhesion. However, further extending the electro-oxidation time would lead to the formation of intramolecular crosslinking in alginate-dopa, thus reducing available dopaquinone and dopa groups for surface bonding and drastically decreased the adhesion strength. The proposed mechanism was supported by the HPLC traces of electro-oxidized N-acetyldopamine products (Fig. S3b-e) for a longer electro-oxidation time. The peak corresponds to quinone decreased while that belongs to dimer increased, indicating the conversion of dopaquinone and dimer. We have added the new comments in the revised manuscript. (See the first paragraph on Page 12 in the revised manuscript; See Fig. S3 on Page 7 and Fig. S10 on Page 11 in the revised Supplementary Information)

Revisions:

... The adhesion strength of Electro-Ox tapes first increased with the increase of the oxidation time in 6 h and then decreased gradually afterwards. We inferred that the dopaquinone groups on alginate reached the maximum at the oxidation time of 6 h. Meanwhile, still quite some dopa remained unoxidized and they contributed to the short-term adhesion. However, further extending the electro-oxidation time would lead to the formation of intramolecular crosslinking in alginate-dopa, thus reducing available dopaquinone and dopa groups for surface bonding and drastically decreased the adhesion strength. The proposed mechanism was supported by the HPLC traces of electro-oxidized N-acetyldopamine products (Fig. S3b-e) at different electro-oxidation time. The peak corresponds to quinone decreased while that belongs to dimer increased, indicating the conversion of dopaquinone and dimer. ...

Fig. S3 Schematic and characterization of the electro-oxidation of N-acetyldopamine. N-acetyldopamine was chosen as the model molecule and oxidized using different methods to confirm the products of electro- and chemical-oxidation. **a**, Different oxidation reaction pathway of dopa. The production of dimer is useless for the surface adhesion of dopa contained materials. In contrast, the production of dopaquinone would extremely benefit the tissue adhesion due to the covalent junction with amino. **b**, HPLC and mass analyses of N-acetyldopamine and the electro- or chemical-oxidation products of N-acetyldopamine. In the electro-oxidation (1 h), most of the phenolic hydroxyl groups changed into quinone while those in the chemical-oxidation (1 h) formed dimer. After further electro-oxidation of N-acetyldopamine by extending the electro-oxidation time (2 h), the quinone in the products would gradually converted to dimer due to the increased dopaquinone concentration. **c-e**, Zoomed-in HPLC curves of the dimer (**c**), dopa (**d**) and quinone (**e**) in **b**. **f**, HPLC and mass analyses of the electro- or chemical-oxidation products of

N-acetyldopamine after storage for 24 h. The remained dopa in the products of chemical-oxidation changed into dimer because of the remanent oxidant while the electro-oxidized products slightly changed, indicating the stability and controllability of the electro-oxidation. **g-i**, Zoomed-in HPLC curves of the dimer (**g**), dopa (**h**) and quinone (**i**) in **f**.

Fig. S10 Optimization of adhesion strength by varying the concentration and the electro-oxidation time of alginate-dopa. **a**, Shear strength of wet porcine skin adhered using the hydrogel tape containing various concentrations of electro-oxidized alginate-dopa (5, 7.5, 10, and 12.5 w/v%). **b**, Shear strength of wet porcine skin adhered using the hydrogel tape containing alginate-dopa electro-oxidized for different times (2, 4, 6, 8, and 24 h). Values represent the mean and standard deviation (n=4-5).

7. In Figure 9, the authors mention using fresh organs, but it appears like the organs have been patted down dry (no blood, bodily fluids; something similar to Figure S11) before applying the adhesive hydrogel. If that is the case, then specific protocols should be mentioned in this case, and appropriate terminology should be used.

Response:

We thank the reviewer for the comments. The word “fresh organs” was not appropriate. We meant that the organs were freshly obtained. After being obtained, the organs were washed with PBS (10 mM, pH=7.4) and patted down gently before applying the hydrogel tapes in the in vitro test. To prevent the dehydration of the organs, PBS (10 mM, pH=7.4) was sprayed onto the organs frequently to keep the surface of the organs wet. Now we have corrected the terminology and added the experimental details in the revised manuscript. (See the third paragraph on Page 28 and the second paragraph on Page 31 in the revised manuscript)

Revisions:

...The fat under the porcine skins was removed with blade and the dermal surface of the porcine skin is cleaned with alcohol and gauze before the experiments....

...The freshly obtained organs were washed with PBS (10 mM, pH=7.4) and patted down gently before applying the hydrogel tapes in the in vitro test. To prevent the dehydration of the organs, PBS (10 mM, pH=7.4) was sprayed onto the organs frequently to keep the surface of the organs wet. ...

Minor comments:

- Line 145 typo: ‘and’
- Figure 3: ‘Shear’ is misspelled again on the Y-axes of (c), (g), (h), and (i).

Response:

We have corrected these typos in the revised manuscript. (See the second paragraph on Page 8 and Figure 3 on Page 17 in the revised manuscript)

Revisions:

Fig. 3 Adhesion performance of the Electro-Ox hydrogel tape. **a**, Cyclic compression and fracture curves versus displacement of instant adhesion for porcine skin using Electro-Ox hydrogel tape in 20 cycles. The top inset corresponds to a schematic of the compression and fracture cycles. The bottom inset corresponds to the magnified fracture curves. **b**, Cyclic compression and fracture curves versus time (top) and normalized tensile strength (ϵ , bottom) of instant adhesion for porcine skin using Electro-Ox hydrogel tape in 20 cycles. Values represent the mean and standard deviation ($n = 3$). **c**, Typical lap shear curves of instant adhesion (0 min), adhesion after the fracture of initial adhesion (0 min'), adhesion after curing for 20 min (20 min) and instant adhesion after the fracture of 20 min adhesion (20 min'). The inset corresponds to the normalized tensile strength of a second instant adhesion event after different times (normalized 0 min', 5 min', 10 min', 20 min', 40 min' and 80 min'). Values represent the mean and standard deviation ($n = 3$). **d**, Image and schematic for

the measurement of shear strength based on the standard lap shear test. *F*, force; *W*, width; *L*, length. **e**, Typical force-displacement curve recorded in the lap shear test and the determination of shear strength. **f**, Shear strength, tensile strength and interfacial toughness of long-term adhesion for different hydrogels. **g**, **h**, Shear strength of short-term (**g**) and long-term (**h**) adhesion for different porcine organs using Electro-Ox hydrogel tape. **i**, Shear strength of long-term adhesion for different substrates (Fe, SiO₂, PMMA and glass) using Electro-Ox hydrogel tape. Insets of **g**, **h** and **i** correspond to the typical force-displacement curves recorded in the lap shear tests. Values represent the mean and standard deviation (n=5). **: p < 0.01, *: p < 0.05.

Reviewer #3 (Remarks to the Author):

This study developed an BSA-Alginate-dopa-PAA hydrogel tape via a controllable electrical oxidation. Owing to the electrical oxidized catecholquinones on the hydrogel, the hydrogel can repeatedly stick to tissue surface initially, and form strong adhesion slowly after hours. This hydrogel tape exhibited strong wet adhesion, including the shear strength, tensile strength and interfacial toughness. The author also demonstrated different potential applications of the BSA-Alginate-dopa-PAA hydrogel. The molecular design of the hydrogel is interesting. However, the advantage of the hydrogel is not impressive enough, which is needed to be promoted.

Response:

We thank the reviewer for the critical comments and suggestions. Following his/her suggestions, we have highlighted the major advances of our work over previous publications in the revised manuscript. We have illustrated the key design principle of fault tolerant hydrogels (See the second paragraph on Page 6 and Fig. 1b on Page 7 in the revised manuscript) and discussed the mechanism underlying different products from chemical and electrical oxidation. We also addressed all the specific comments raised by the reviewer. Hope he/she will find the revised manuscript acceptable for publication in Nature Communications.

Revisions:

...Due to the slow reaction of dopaquinone and amine or thiol on the tissue surfaces, we expect the hydrogel tapes can provide long-term strong adhesion as chemical bonding hydrogel tapes yet have a time window for reversible detachment, thus combine the merits of both physical and chemical bonding-based hydrogel tapes (Fig. 1b). ...

Fig. 1b Comparison of the time-dependent adhesion strength of common hydrogel tapes using either physical bonding or chemical bonding for surface adhesion and the fault tolerant hydrogel tapes using a time-dependent formation of covalent bonding for surface adhesion. The overall adhesion strength (W_{adhesion} , solid line) depends on the lower one between the break strength of the hydrogel (W_{hydrogel} , dash line) and the surface bonding energy (W_{surface} , dash line). Physical surface bonding allows to detach the hydrogels freely but cannot ensure strong adhesion. In contrast, chemical surface bonding ensures strong adhesion but is not detachable. Due to the slow formation of chemical bonds with surface, the fault tolerant hydrogel tapes can provide strong adhesion as chemical bonding hydrogel tapes yet have a time window for reversible detachment, thus combine the merits of both physical and chemical bonding-based hydrogel tapes.

Specific comments:

1) The dopa can be oxidized in air. During the synthesis process of alginate-dopa, how to prevent the catechols from oxidation.

Response:

We thank the reviewer for the comments. We have now provided the synthetic details in the revised manuscript. During the synthesis, to avoid the oxidation of dopa in air, ascorbic acid was added to the solution and the solution was degassed. The whole synthesis and dialysis process of alginate-dopa was performed under the protection of argon. Now we have included these experimental details in the revised manuscript. (See the first paragraph on Page 3 in the revised Supplementary Information)

Revisions:

...Typically, alginate and dopa were dissolved in ddH₂O to concentrations of 0.25 mM and 0.1 M, respectively. Then, EDC and NHS were added to the solution to a concentration of 0.6 M. Ascorbic acid was added into the mixture to the concentration of 0.3 M and the pH was adjusted to 7.8 using NaOH solution. Then the mixture was degassed with argon and sonicated three times (each time for 15 min) to remove dissolved oxygen. Ascorbic acid and degassing were used to prevent the oxidation of dopa. The mixture was stirred for 12 h at room temperature under the protection of argon. Finally, the unreacted reactants were removed by dialysis in ddH₂O under the protection of argon, and the product was lyophilized. ...

2) Excitingly, as a hydrogel, the shear and tensile strength of the Electro-Ox hydrogel reached to MPa, which are higher than that of catechol-based hydrogel and catechol-based adhesives or glues, especially under wet condition. It is interesting, please give more discussion on it.

Response:

We thank the reviewer for the comments. The high shear and tensile strength of the Electro-Ox hydrogel comes from the unique crosslinking mechanism and the network structures. In the Electro-Ox hydrogel, the covalent crosslinks include the amide bond between the amino groups on the BSA surface and the carboxyl groups of alginates

and/or PAA, and some dopaquinone-amine adducts. The physical crosslinks include the hydrogen bonding between dopa and dopaquinone¹², π - π stacking among dopa groups¹³, the charge-charge interaction between BSA and PAA/alginate, and numerous physical interactions in the folded BSA structure. The rupture of the physical interactions prior to the break of covalent bonds can efficiently dissipate energy. Moreover, BSA is 7.1 nm in diameter and serve as a crosslinking hub. Similar to the nanoparticle-crosslinked hydrogels¹⁴, the nanosized BSA can prevent crack propagation and further increase the tensile strength. Moreover, the formation of abundant dopaquinone and amine/thiol covalent bonds ensures long-term strong surface bonding. The combined strong surface bonding and the strength of the hydrogel matrix ensure the overall shear and tensile strength of the hydrogel tapes over megapascal. Now we have discussed the mechanism in the revised manuscript. (See the second paragraph on Page 12 and line 260-268 on Page 13 in the revised manuscript).

Revisions:

...The excellent adhesion strength mainly attributed to the strong adhesion and cohesion of the hydrogels. The outstanding shear and tensile strength of the Electro-Ox hydrogel comes from the unique crosslinking mechanism and the network structures. In the Electro-Ox hydrogel, the covalent crosslinks include the amide bond between the amino groups on the BSA surface and the carboxyl groups of alginates and/or PAA, and some dopaquinone-amine adducts. The physical crosslinks include the hydrogen bonding between dopa and dopaquinone⁴², π - π stacking among dopa groups³⁹, the charge-charge interaction between BSA and PAA/alginate, and numerous physical interactions in the folded BSA structure. The rupture of the physical interactions prior to the break of covalent bonds can efficiently dissipate energy. Moreover, BSA is 7.1 nm in diameter and serve as a crosslinking hub. Similar to the nanoparticle-crosslinked hydrogels⁶⁹, the nanosized BSA can prevent crack propagation and further increase the tensile strength. Moreover, the formation of abundant dopaquinone and amine/thiol covalent bonds ensures long-term strong surface bonding. The combined strong surface bonding and the strength of the hydrogel matrix ensure the overall shear and tensile strength of the hydrogel tapes over megapascal....

3) Breaking the interfacial hydration layer is critical for wet adhesion. Without the interfacial hydration layer, the adhesive groups can bond with the substrate. In Xuanhe Zhao's work, the PAA-based path is dried, thus their path can adsorb the interfacial water and bond with the substrate. However, in this work, the Electro-Ox hydrogel contained water, which is hard to adsorb interfacial water or break the hydration layer. Thus, the mechanism of wet adhesion in this work should be clearly demonstrated.

Response:

We thank the reviewer for the comments. As indicated by the reviewer, breaking the interfacial hydration layer is critical for wet adhesion. Similar as that of Zhao's work, the Electro-Ox hydrogel can also absorb the interfacial water and quickly swell at the surface. We believe that the ability of a hydrogel to remove the surface water depends not only on how much but also on how fast it can absorb surface water. Our hydrogels

can absorb water of ~ 1.5 - 2.2 times the original weight of the hydrogel (Fig. S5). This is sufficient for absorbing the water in the surface hydration layer. To illustrate how fast the hydrogel tape can absorb water, we monitored the hydration of an Electro-Ox hydrogel in PBS. As shown in Figure S11, the hydrogel absorbed water quickly and the thickness increased 25% in less than 40 s. Assuming the hydrogel swells uniformly in three dimensions, the absorbed water is ~ 0.6 time of its own weight. Quickly swelling at the interface allows breaking the interfacial hydration layer and ensures the strong short-term bonding of Electro-Ox hydrogel. Now we have included the new comments and data in the revised manuscript. (See line 274-276 on Page 13 in the revised manuscript; See Fig. S11 on Page 11 in the revised Supplementary Information).

Revisions:

...It is worth mentioning that Electro-Ox hydrogels can quickly absorb the interfacial water to break the interfacial hydration layer, leading to the direct contact of the adhesive groups with the substrates (Fig. S11). ...

Fig. S11 Swelling of the Electro-Ox hydrogel in PBS solution (10 mM, pH=7.4). **a**, Microscopic images of the cross section of the Electro-Ox hydrogel after soaking the hydrogel in PBS solutions for different times. **b**, Summarized variation of the thickness of hydrogels with time.

4) The study demonstrated that the BSA-alginate-dopa-PAA hydrogel tape is biodegradable. However, most of the components in the hydrogel is PAA, which is not degradable.

Response:

We thank the reviewer for the comments. As the reviewer pointed out, PAA and its derivatives are not completely degradable under physiological conditions¹⁵, although it is biocompatible in many biomedical applications as reported in literature^{1, 16-18}. PAA is approved by FDA as a stabilizer and thickener in defoaming agents. We used “biodegradable” to indicate that the hydrogel tape can be decomposed in vivo, which is supported by our experiments shown in Fig. 6c and d. We think the degradation of

the hydrogels is mainly due to the enzymatic digestion of BSA¹⁹, which leads to the decomposition of the hydrogel network. We have emphasized this in the revised manuscript. (See the first paragraph on Page 22 in the revised manuscript).

Revisions:

...Because PAA and its derivatives are not degradable under physiological conditions⁷⁰ and the degradation of alginate in vivo is slow and uncontrollable⁷¹, we think the degradation of the hydrogels is mainly due to the enzymatic digestion of BSA, which leads to the decomposition of the hydrogel network. ...

5) The advantage design of the study is that with the controllable oxidization, the hydrogel can repeat adhere at the initial time, and form strong adhesion slowly. Compare with the rapid surface covalent bonding, this design can avoid the incorrect adhesion. However, clinically, most applicants require initial strong adhesion. The dynamic environment in body, such as the heartbeat, does not allow slow adhesion on the heart's surface for hours. Moreover, the potential applications in this study are also used the initial adhesion. Thus, the importance of the strong adhesion in the second stage needs to be reconsidered.

Response:

We thank the reviewer for this critical comment. Indeed, most applications require initial strong adhesion strength as the reviewer pointed out. As shown in Fig. 3g, we evaluated the short-term wet adhesion strength of the Electro-Ox hydrogel tapes for different organic tissues such as vessel, stomach, liver, intestine and heart. The initial adhesion strength of all the tissues was higher than 30 kPa (~78 kPa for vessel, ~31 kPa for stomach, ~32 kPa for liver, ~33 kPa for intestine, and ~40 kPa for heart), which is strong enough for the instant adhesions. In our case, the initial strong adhesion is based on physical interactions. These interactions ensure reversible attachment and detachment of the hydrogel tapes. However, these interactions are severely weakened due to swelling of the adhesives or bleeding from the tissue²⁰⁻²⁴. Therefore, slow formation of strong covalent bonds is critical for the potential applications such as wound closure, fixing implantable devices, and haemostasis. As such we think the strong adhesion in the second stage is also important. Now we have commented on this in the revised manuscript. (See the end of Page 15 and the first paragraph on Page 16 in the revised manuscript).

Revisions:

...The short-term adhesion of the Electro-Ox hydrogel tapes on different tissues were larger than 30 kPa, which is strong enough for instant tissue adhesions. The initial strong adhesion is based on physical interactions. These interactions ensure reversible attachment and detachment of the hydrogel tapes. However, these interactions are severely weakened due to swelling of the adhesives or bleeding from the tissue²⁴⁻²⁸. Therefore, slow formation of strong covalent bonds is critical for the potential applications such as wound closure, fixing implantable devices, and haemostasis. The

formation of covalent bonding between dopaquinone and amino/thiol increased the adhesion strength to more than 100 kPa, making the hydrogel tapes suitable for long-term applications. ...

6) The wet adhesion of hydrogels is most used for biomedical applications. How much adhesion can meet biomedical applications? In the study, the adhesion of the Chem-Ox and non-OX hydrogel are higher than that of many reported hydrogels. What are their disadvantages in biomedical applications? For examples, can these hydrogels be used in the same applications in this study?

Response:

We thank the reviewer for the comments. The normal blood pressure and intrapulmonic pressure of human were less than 18.6 and 18.7 kPa. The adhesion strength (Shear strength) of most of the commercial adhesives was less than 100 kPa according to the previous report (Zhao, et al. *Nature* **575**, 169–174 (2019)). So, we anticipate that the adhesion strength suitable for biomedical applications are in the range of 20-100 kPa, which is smaller than that of the Electro-Ox hydrogel tapes reported in this manuscript. Even though the Chem-Ox and non-Ox hydrogels also exhibited considerable adhesion strength, they are limited for certain applications. For non-Ox hydrogels, the adhesion strength to organic tissues is relatively weak compared to Chem-Ox and Electro-Ox hydrogels as shown in Figure 3f. For Chem-Ox hydrogels, the uncontrollable oxidation of dopa (Fig. S3) and the generation of dimer would affect the adhesion performance on inorganic substrates compared to Non-Ox and Electro-Ox hydrogels. As a result, we anticipate these hydrogels cannot be used for the same applications in this study. Now we have commented on this in the revised manuscript. (See line 268-274 on Page 13 in the revised manuscript).

Revisions:

...Even though the Chem-Ox and Non-Ox hydrogels also exhibited considerable adhesion strength, they are limited for certain applications. For non-Ox hydrogels, the adhesion strength to organic tissues is relatively weak compared to Chem-Ox and Electro-Ox hydrogels as shown in Fig. 3f. For Chem-Ox hydrogels, the uncontrollable oxidation of dopa (Fig. S3) and the generation of dimer would affect the adhesion performance on inorganic substrates compared to Non-Ox and Electro-Ox hydrogels. ...

References:

1. Yuk H, *et al.* Dry double-sided tape for adhesion of wet tissues and devices. *Nature* **575**, 169-174 (2019).
2. Akram Bhuiyan MS, *et al.* In Situ Deactivation of Catechol-Containing Adhesive Using Electrochemistry. *J. Am. Chem. Soc.* **142**, 4631-4638 (2020).
3. Lakshmi D, *et al.* Electrochemical Sensor for Catechol and Dopamine Based on a Catalytic Molecularly Imprinted Polymer-Conducting Polymer Hybrid Recognition Element. *Anal. Chem.* **81**, 3576-3584 (2009).

4. Mao A, Li H, Jin D, Yu L, Hu X. Fabrication of electrochemical sensor for paracetamol based on multi-walled carbon nanotubes and chitosan–copper complex by self-assembly technique. *Talanta* **144**, 252-257 (2015).
5. Brubaker CE, Messersmith PB. Enzymatically Degradable Mussel-Inspired Adhesive Hydrogel. *Biomacromolecules* **12**, 4326-4334 (2011).
6. Guvendiren M, Messersmith PB, Shull KR. Self-Assembly and Adhesion of DOPA-Modified Methacrylic Triblock Hydrogels. *Biomacromolecules* **9**, 122-128 (2008).
7. Lee BP, Dalsin JL, Messersmith PB. Synthesis and Gelation of DOPA-Modified Poly(ethylene glycol) Hydrogels. *Biomacromolecules* **3**, 1038-1047 (2002).
8. Kim BJ, *et al.* Mussel-mimetic protein-based adhesive hydrogel. *Biomacromolecules* **15**, 1579-1585 (2014).
9. Lee BP, Messersmith PB, Israelachvili JN, Waite JH. Mussel-Inspired Adhesives and Coatings. *Annu. Rev. Mater. Res.* **41**, 99-132 (2011).
10. Shao C, *et al.* Mussel-Inspired Cellulose Nanocomposite Tough Hydrogels with Synergistic Self-Healing, Adhesive, and Strain-Sensitive Properties. *Chem. Mater.* **30**, 3110-3121 (2018).
11. Han L, *et al.* Mussel-Inspired Adhesive and Conductive Hydrogel with Long-Lasting Moisture and Extreme Temperature Tolerance. *Adv. Funct. Mater.* **28**, 1704195 (2018).
12. Ahn BK, Lee DW, Israelachvili JN, Waite JH. Surface-initiated self-healing of polymers in aqueous media. *Nat. Mater.* **13**, 867-872 (2014).
13. Li Y, Cao Y. The molecular mechanisms underlying mussel adhesion. *Nanoscale Adv.* **1**, 4246-4257 (2019).
14. Thoniyot P, Tan MJ, Karim AA, Young DJ, Loh XJ. Nanoparticle–Hydrogel Composites: Concept, Design, and Applications of These Promising, Multi-Functional Materials. *Adv. Sci.* **2**, 1400010 (2015).
15. Lee KY, Mooney DJ. Hydrogels for Tissue Engineering. *Chem. Rev.* **101**, 1869-1880 (2001).
16. Xu C, *et al.* Biodegradable Nanoparticles of Polyacrylic Acid–Stabilized Amorphous CaCO₃ for Tunable pH-Responsive Drug Delivery and Enhanced Tumor Inhibition. *Adv. Funct. Mater.* **29**, 1808146 (2019).
17. Hu X, *et al.* Preparation and characterization of a novel pH-sensitive Salecan-g-poly(acrylic acid) hydrogel for controlled release of doxorubicin. *J. Mater. Chem. B* **3**, 2685-2697 (2015).
18. Jing Z, Xu A, Liang YQ, Zhang Z, Li Y. Biodegradable Poly(acrylic acid-co-acrylamide)/Poly(vinyl alcohol) Double Network Hydrogels with Tunable Mechanics and High Self-healing Performance. *Polymers* **11**, 952 (2019).
19. Ma X, *et al.* A Biocompatible and Biodegradable Protein Hydrogel with Green and Red Autofluorescence: Preparation, Characterization and In Vivo Biodegradation Tracking and Modeling. *Sci. Rep.* **6**, 19370 (2016).
20. Zhao Y, *et al.* Bio-inspired reversible underwater adhesive. *Nat. Commun.* **8**, 2218 (2017).
21. Bradley LC, Bade ND, Mariani LM, Turner KT, Lee D, Stebe KJ. Rough Adhesive Hydrogels (RAD gels) for Underwater Adhesion. *ACS. Appl. Mater. Inter.* **9**, 27409-

27413 (2017).

22. Fan H, *et al.* Adjacent cationic–aromatic sequences yield strong electrostatic adhesion of hydrogels in seawater. *Nat. Commun.* **10**, 5127 (2019).

23. Han L, Wang M, Prieto-López LO, Deng X, Cui J. Self-Hydrophobization in a Dynamic Hydrogel for Creating Nonspecific Repeatable Underwater Adhesion. *Adv. Funct. Mater.* **30**, 1907064 (2020).

24. Balkenende DWR, Winkler SM, Li Y, Messersmith PB. Supramolecular Cross-Links in Mussel-Inspired Tissue Adhesives. *ACS Macro Lett.* **9**, 1439-1445 (2020).

REVIEWER COMMENTS

Reviewer #1 (Remarks to the Author):

The authors provided a thorough point-by-point response to the questions. Moreover, the authors supplied extensive new data to fill in perceived gaps especially with regards to certain control groups and added time points. The authors provided supportive evidence with regards to the adhesive's performance in various anatomical/physiological scenarios, reversibility, and with some caveats biocompatibility. As a result, the manuscript has been significantly improved. The presentation of the technology appears overall more well rounded.

There may have been some confusion regarding one of the comments:

Re biocompatibility - the authors provide H&E imaging of remote organs after subcutaneous implantation of their bioadhesive in the back of rats. However, the authors did not test the longterm effects of the bioadhesive when directly applied on said target organs for an extended period of time as suggested in the first review (see quote below).

"9. More importantly the authors only provide in vivo data of subcutaneous implantation, but no data on the histologic response of the proposed target organs (stomach, lung, artery) to the chronic tape administration."

Unless I misread the S27/28 figure legends, the bioadhesive was implanted into the back of the rats and then the remote organs were harvested for analysis.

Also, the analysis only consisted of H&E staining and therefore is very limited with regards to any immune/healing response (CD45, macrophage markers, collagen)

Reviewer #2 (Remarks to the Author):

My comments have been addressed in great detail, thanks!

Reviewer #3 (Remarks to the Author):

The authors have addressed my concerns. It is a great job. I would like to recommend the manuscript for publication in Nature communications as is.

Chaoming Xie

Point-by-point response to the reviewers' comments

Reviewer #1 (Remarks to the Author):

The authors provided a thorough point-by-point response to the questions. Moreover, the authors supplied extensive new data to fill in perceived gaps especially with regards to certain control groups and added time points. The authors provided supportive evidence with regards to the adhesive's performance in various anatomical/physiological scenarios, reversibility, and with some caveats biocompatibility. As a result, the manuscript has been significantly improved. The presentation of the technology appears overall more well rounded.

Response:

We thank the reviewer for his/her positive comments on our revisions.

There may have been some confusion regarding one of the comments:

Re biocompatibility - the authors provide H&E imaging of remote organs after subcutaneous implantation of their bioadhesive in the back of rats. However, the authors did not test the longterm effects of the bioadhesive when directly applied on said target organs for an extended period of time as suggested in the first review (see quote below).

"9. More importantly the authors only provide in vivo data of subcutaneous implantation, but no data on the histologic response of the proposed target organs (stomach, lung, artery) to the chronic tape administration."

Unless I misread the S27/28 figure legends, the bioadhesive was implanted into the back of the rats and then the remote organs were harvested for analysis.

Also, the analysis only consisted of H&E staining and therefore is very limited with regards to any immune/healing response (CD45, macrophage markers, collagen)

Response:

We are a bit confused with this comment. We thought that in the question quoted from the first round of review, the reviewer asked for studying the histologic response of the proposed target organs to the chronic tape administration instead of studying the immune/healing response.

Similarly, in a few recent reports of bio-adhesives, the in vivo biocompatibility was also evaluated in mouse/rat models by the dorsal subcutaneous implantation (*Nature* **575**, 169-174 (2019); *Nat. Mater.* **20**, 229–236 (2021); *Adv. Funct. Mater.* **30**, 1907064 (2020)) and the histologic analysis, including H&E staining and immunofluorescent staining. Beyond these analyses, in our work we adapted the protocol from another recent publication (*Nat. Commun.* **12**, 1689 (2021)) and performed H&E staining and blood analysis to indicate the biosafety of the hydrogel to major organs and blood at Day 1, 7, and 14 after subcutaneous implantation of hydrogels. We think these evaluations and results may be sufficient to support the biocompatibility of the hydrogels at the laboratory level (Fig. 6 and Supplementary Fig. 24-28).

We do share the same concern as this reviewer on the immune response of our hydrogel tapes for real clinic applications. Certainly, more rigorous biocompatibility test should be conducted prior to the use the hydrogel tapes to human bodies. For this purpose, the concerns

about the immune response under different conditions cannot be exhaustively addressed by the experiments proposed by the reviewer. Using a single animal model to study the immune response of biomedical devices is often insufficient.

On the other hand, the novelty of this work lies in the creation of a fault tolerant hydrogel tapes containing electrically produced dopaquinone suitable for strong wet adhesion. The polymers and chemicals we used in the hydrogels are all proved by FDA. Including additional experiments about the host immune/healing response of our materials may just have limited benefit to the novelty of this paper.

Therefore, we hope the reviewer can agree with us to tune down our claim on the biocompatibility of our hydrogel tapes in the revised manuscript instead of providing further experiments. We have included the following statement in the revised manuscript:

“Currently, we did not study the immune/healing response of the target tissues (stomach, lung, and artery) in direct contact to the hydrogel tapes. Although the ex-vivo experiments show the possible applications of the Electro-Ox hydrogel tapes on these organs, we realize that the long-term biocompatibility and host immune/healing response should be more rigorously investigated before the clinic applications of the hydrogel tapes.” (See the second paragraph on Page 24 in the revised manuscript)

Reviewer #2 (Remarks to the Author):

My comments have been addressed in great detail, thanks!

Response:

We thank the reviewer for the comments.

Reviewer #3 (Remarks to the Author):

The authors have addressed my concerns. It is a great job. I would like to recommend the manuscript for publication in Nature communications as is.
Chaoming Xie

Response:

We thank the reviewer for the comments.